# Quantifying collective interactions in biomolecular phase separation

Hannes Ausserwöger [1,9], Ella de Csilléry [1,9], Daoyuan Qian [1,9], Georg Krainer[2,9], Timothy J. Welsh [1], Tomas Sneideris [1], Titus M. Franzmann [3], Seema Qamar[4], Nadia A. Erkamp [1], Jonathon Nixon-Abell [4], Mrityunjoy Kar [5], Peter St George-Hyslop[6,7], Anthony A. Hyman [5], Simon Alberti [3], Rohit V. Pappu [8] & Tuomas P. J. Knowles [1] ✉

Biomolecular phase separation is an emerging theme for protein assembly and cellular organisation. The collective forces driving such condensation, however, remain challenging to characterise. Here we show that tracking the dilute phase concentration of only one component suffices to quantify composition and energetics of multicomponent condensates. Applying this assay to several disease- and stress-related proteins, we find that monovalent ions can either deplete from or enrich within the dense phase in a context-dependent manner. By analysing the effect of the widely used modulator 1,6-hexanediol, we find that the compound inhibits phase separation by acting as a solvation agent that expands polypeptide chains. Extending the strategy to *in cellulo* data, we even quantify the relative energetic contributions of individual proteins within complex condensates. Together, our approach provides a generic and broadly applicable tool for dissecting the forces governing biomolecular condensation and guiding the rational modulation of condensate behaviour.

The assembly of proteins is a ubiquitous phenomenon in living systems that enables essential functions spanning from synthesis[1,2] to cellular signalling[3,4]. Recently, phase separation of proteins into biomolecular condensates has been discovered as a pathway for protein assembly[5–7]. Through phase separation, proteins spontaneously demix from a homogenous phase into a protein-rich, condensed phase and a protein-poor, dilute phase[8,9]. Such condensate formation has been attributed an important role in cellular organisation[10–13] and is associated with aberrant behaviour in pathology[14].

Biomolecular phase separation is driven by collective interactions, stemming from the collaborative association of large numbers of molecules. These collective interactions give rise to emergent behaviours such as differential partitioning of solutes between dense and dilute phases. Conversely, individual components also differ in their contributions to the overall decrease in free energy, acting as the driving force for phase separation. Common protein interaction assays, however, fail to inform of the collective properties underlying phase separation, given a focus on binary interactions. Even assays that

[1]Centre for Misfolding Diseases, Yusuf Hamied Department of Chemistry, University of Cambridge, Cambridge, UK. [2]Institute of Molecular Biosciences (IMB), University of Graz, Graz, Austria. [3]Biotechnology Center (BIOTEC), Center for Molecular and Cellular Bioengineering (CMCB), Technische Universität Dresden, Dresden, Germany. [4]Cambridge Institute for Medical Research, Department of Clinical Neurosciences, Clinical School, University of Cambridge, Cambridge, UK. [5]Max Planck Institute of Cell Biology and Genetics (MPI-CBG), Dresden, Germany. [6]Department of Medicine (Division of Neurology), Temerty Faculty of Medicine, University Health Network, University of Toronto, Toronto, ON, Canada. [7]Department of Neurology, Columbia University, New York, NY, USA. [8]Department of Biomedical Engineering and Center for Biomolecular Condensates, Washington University in St. Louis, St. Louis, MO, USA. [9]These authors contributed equally: Hannes Ausserwöger, Ella de Csilléry, Daoyuan Qian, Georg Krainer. ✉e-mail: tpjk2@cam.ac.uk

map out the phase boundary, which refers to the set of critical concentrations[15–17], do not provide information on the dense phase composition and fail to quantify the energetics.

Recent theoretical advances have introduced novel descriptors that enable the quantification of collective interactions[18,19]. These descriptors are derived from the analysis of tie line gradients[18] and the quantification of component dominance[19], and provide means to physically characterise the mechanisms of condensate formation and modulation. Tie lines describe the demixing process between biomolecular dense and dilute phases, providing insights into condensate composition and the underlying interactions. Additionally, component dominance analysis offers the opportunity to discern individual components' contributions to the decrease in free energy, thus shedding light on the underlying driving forces and energetics in a locally defined manner. Hence, translating these theoretical advances into commonly accessible experiments addresses the current limitations in characterising the collective interactions underlying protein phase separation.

Here, we establish a general strategy for quantifying the collective interactions in biomolecular phase separation with a widely accessible experimental approach. By applying this framework, we shed light on condensate composition and the underlying energetics of a range of physiologically relevant phase separation-prone proteins. We find that even elemental ions commonly display concentration gradients between the dense and dilute phases, and that partitioning effects are context-dependent on the governing interactions. We further show that the common small-molecule hydrophobic disruptor 1,6-hexanediol acts by regulating polypeptide chain expansion and even highlight how our approach can be applied intracellularly.

## Results

### Descriptors for quantifying collective interactions

Tie lines connect the equilibrium dilute and dense phase concentrations of constituent species formed after demixing, and so describe condensate composition including partitioning (see Fig. 1a).

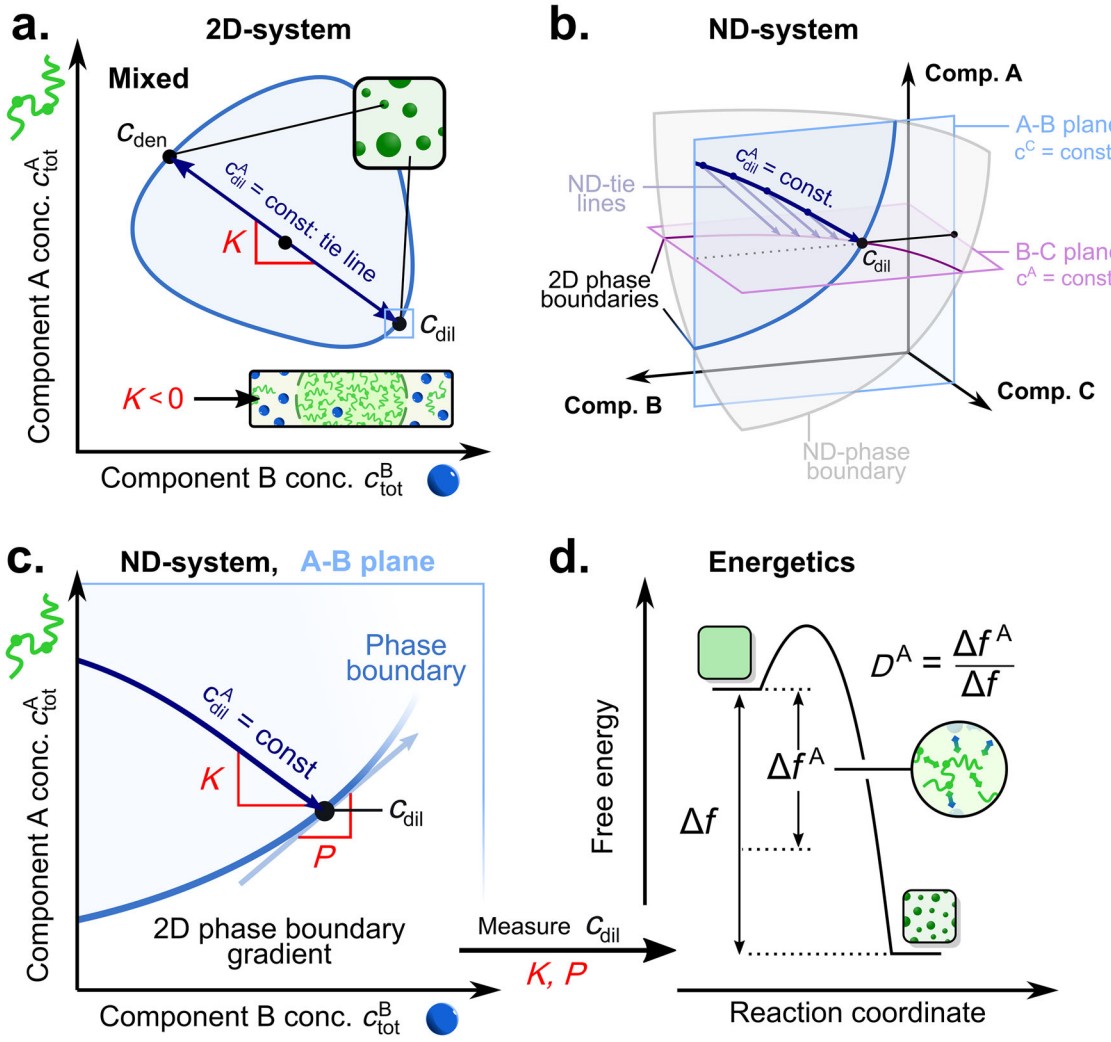

**Fig. 1 | Descriptors of collective interactions in biomolecular phase separation.** **a** Schematic of a full phase diagram for a 2-D system between components A and B (abbreviated as Comp.). The coexistence region is shaded in blue. Tie lines describe the demixing of a total composition ($c_{tot}$) within the coexistence region into dilute ($c_{dil}$) and dense phase concentrations ($c_{den}$). The tie line gradient $K$ is informative of component partitioning and can be obtained by determining conditions of $c^A_{dil}$ = const. **b** In higher-dimensional systems, tie lines do not have to be constrained to the A-B measurement plane (blue). Enforcing the condition $c^A_{dil}$ = const., reduces the higher-dimensional tie lines to the A-B measurement plane, resulting in a so-called dilute phase contour. Dilute phase contours correspond to the intersection of the higher-dimensional tie lines originating from the B-C plane at $c^A_{dil}$ = const (purple), with the A-B measurement plane of interest. **c** Reduction of tie lines to the A-B measurement plane gives rise to information on $K$. Furthermore, the local dependence of the saturation concentration on both component A and B is described by the phase boundary gradient $P$. **d** Together $K$ and $P$ inform on the relative free energy decrease for phase separation of component A as given by the dominance of component A ($D^A$). Both $K$ and $P$ can be determined from measurements of $c^A_{dil}$ only.

In a two-component system, the tie line gradient can be determined by identifying a set of total concentration conditions that fulfil $c_{dil}^A$ = const. For higher-dimensional systems, as is generally the case in practical settings (additional components include ions, buffers, etc.), tie lines can be approximated by the same approach of identifying the contour $c_{dil}^A$ = const[18,19]. This leads to a dimensionality reduction of tie lines to the A-B measurement plane of interest, by intersection of the higher-dimensional tie lines with said plane (Fig. 1b, Supplementary Fig. 1). Crucially, the reduced tie line gradient $K$ preserves essential information on the partitioning of (co-)solutes of interest as it approximates the ratio of the higher-dimensional tie line gradient entries in A and B[19]. Specifically, for $K < 0$, component A is enriched in the dense phase (i.e., included in condensates), while component B is decreased (i.e. excluded from condensates). By contrast, a $K > 0$ means that both species are present at higher concentrations in the dense phase than in the dilute phase, indicating preferential partitioning into condensates.

In addition to tie lines, the shape of the phase boundary plays an essential role in understanding collective interactions. The phase boundary describes the local dependence of the saturation concentration (Fig. 1c) on the individual components of interest. This relationship can be quantified by the local phase boundary gradient $P$. In the extreme cases where $P$ approaches 0 or $\infty$, the onset of phase separation becomes solely dependent on component B or A, respectively. Together, $P$ and $K$ inform on the energetics of the phase separation process according to (Fig. 1d, see "Derivation of the dominance framework" section)[19]:

$$D^A = \frac{\Delta f^A}{\Delta f} = \frac{K}{K - P} \tag{1}$$

where $D^A$ denotes the dominance of component A and $\Delta f^A$ the free energy decrease of component A with regard to the overall free energy decrease $\Delta f$[19]. $D^A$ gives the relative energetic contribution of component A to the overall free energy decrease, which provides a measure of the system's energetic dependence on said component.

Reduced tie line gradients $K$, phase boundary gradients $P$ and dominance $D^A$ can be extracted by measuring dilute phase concentrations of one component only, as shown in Fig. 2. Such dilute phase concentration measurements can be performed using image-based confocal detection or even standard epifluorescence or absorbance-based measurements post-separation of dilute and dense phase. Here, we have chosen a microfluidic flow cell setup with a scanning confocal microscope (see Fig. 2a; Supplementary Fig. 2 for confocal measurement setup) to optimise precision and throughput. The dilute phase concentrations are then extracted from the baseline signal in the fluorescence time traces (Fig. 2b, c). Short bursts of intensity represent condensates but are not considered for further analysis.

To extract $K$, $P$ and $D^A$, the change in the dilute phase concentration of component A is quantified by conducting a series of experiments with constant total concentration of component A ($c_{tot}^A$) and varying total concentration of component B ($c_{tot,1}^B$) (see Fig. 2d). These so-called heterotypic line scans are then performed at two different constant total concentrations of component A (i.e., $c_{tot,1}^A$ and $c_{tot,2}^A$). Measurements of additional heterotypic line scans at different concentrations, followed by averaging, can improve the accuracy, but are not necessary. At higher concentrations of component B, the heterotypic line scan displays a flat plateau region because no phase separation occurs in this range. Consequently, the dilute phase concentration is equal to the total concentration. However, at total concentrations of component B below the saturation concentration of component B ($c_{sat}^B$), phase separation takes place, leading to a decrease in the dilute phase concentration due to the formation of condensates.

The phase boundary gradient $P$ is then determined from the change in $c_{sat}^B$ with changing $c_{tot}^A$:

$$P = \frac{c_{tot,2}^A - c_{tot,1}^A}{c_{sat,2}^B - c_{sat,1}^B} = \frac{\Delta c_{tot}^A}{\Delta c_{sat}^B} \tag{2}$$

In order to calculate $K$, the heterotypic dilute phase response gradient $R_{AB}$ is determined from the initial slope of the decrease in dilute phase concentration (Fig. 2d). $R_{AB}$ quantifies the change in dilute phase concentration of component A with increasing concentrations of component B. From $P$ and $R_{AB}$, the reduced tie line gradient $K$ is calculated according to[19]:

$$K = \frac{R_{AB} * P}{R_{AB} - P} \tag{3}$$

Finally, from $K$ and $P$, the dominance $D$ of component A is determined according to Eq. 1.

## Ions are excluded from FUS condensates at low ionic strengths to counteract charge screening

The intrinsically disordered protein FUS is involved in physiologically relevant intracellular condensation events[13,20–22] and associated with the emergence of amyotrophic lateral sclerosis (ALS)[13,23,24]. A hallmark of FUS is its tendency to undergo phase separation with decreasing ionic strengths, but key features such as ion partitioning or the impact of ions on the energetic driving force remain unaddressed.

Hence, we set out to characterise the collective interactions of FUS with salt species using the tie line gradient and dominance analysis approach. We first determined the dilute phase concentrations of FUS at 1 and 2 μM total protein concentration and KCl concentrations ranging from 20 to 150 mM (Fig. 3a, b and Supplementary Fig. 3). At KCl concentrations >75 mM, the protein dilute phase concentration did not vary, as no phase separation was observed (see Fig. 3b). Only with decreasing charge screening at lower KCl concentrations a drop-off in $c_{dil}^{FUS}$ was observed, corresponding to the onset of phase separation. This enables determination of the dilute phase response gradient $R_{AB} = 0.020 \pm 0.003$ μM/mM, which confirms that increasing KCl concentrations cause an increase in the FUS dilute phase concentration. The phase boundary gradient of $P = 30.3 \pm 8.1$ μM/mM highlights that lower concentrations of KCl are required to induce phase separation at lower protein concentrations. This can be further illustrated by determining the FUS/KCl phase boundary in the experimentally probed concentration range (Fig. 3c, Supplementary Fig. 4 for phase boundary determination approach).

We then determined the reduced tie line gradient as $K = -0.05 \pm 0.04$ μM/mM. A negative gradient indicates that the ion concentration is lower in the dense phase than in the dilute phase (Fig. 3c). Here, considering the dissociation of the salt into $K^+$ and $Cl^-$, the obtained gradient corresponds to the ratio of protein to a weighted sum of the cationic and anionic species, with the weights relating to their valency[19]. Hence, on average, the ions are preferentially excluded from FUS condensates, likely causing a decrease in electrostatic shielding and strengthening protein–protein interactions. Such ion exclusion from the dense phase is corroborated by simulation results for FUS condensates[25] and has also been observed for polyelectrolyte polymer phase separation[26,27], while recent evidence further highlights that such ion partitioning and associated charge neutralisation can even drive pH gradients[28].

KCl has further been shown to interact with a large fraction of available polypeptide side chains[29], forming a hydration shell around the dispersed protein, which inhibits intermolecular protein interactions. A release of ions from the hydration shell would therefore allow for FUS to engage in much more effective protein–protein interactions

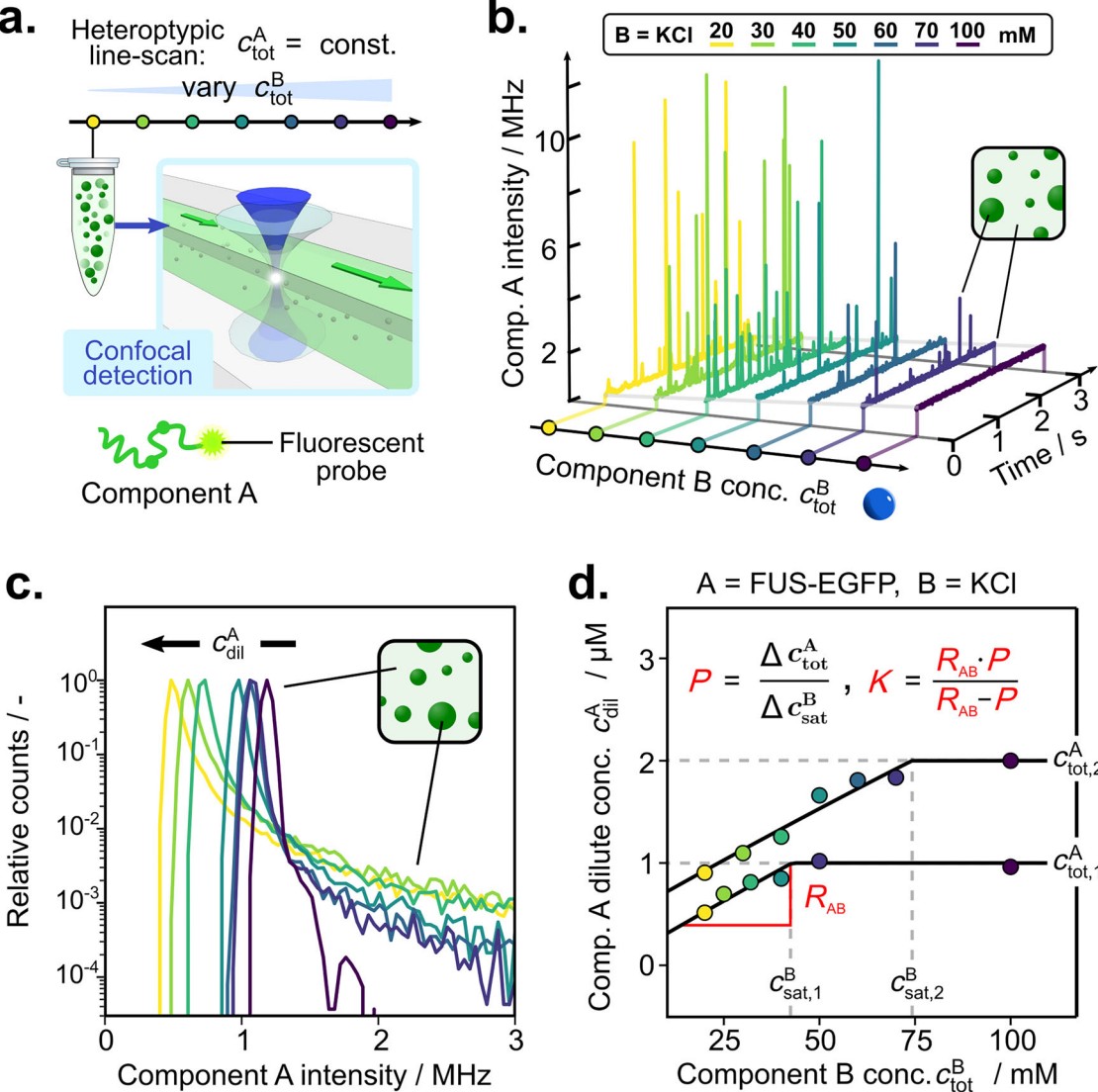

**Fig. 2 | Quantification of collective interactions using one-component dilute phase concentrations measurements. a** Heterotypic line scans are performed by injecting samples with constant concentrations of component A but varying concentrations of component B (illustrated by yellow-purple colour gradient) into a microfluidic flow cell. Readout of fluorescence is done by confocal detection. **b** Representative intensity time traces of the component obtained from confocal detection measurements. **c** Intensity histograms obtained from time traces in (**b**), where the maximum is representative of the dilute phase concentration of

component A. **d** Component A (FUS-EGFP … Fused in sarcoma protein with a enhanced green fluorescent protein tag), dilute phase concentration changes with varying component B (KCl … potassium chloride) concentrations along two separate line scans. The raw data for the line scan at $c_{tot}^{FUS} = 2\,\mu M$ is shown in (**b, c**), and the corresponding data for the line scan at $c_{tot}^{FUS} = 1\,\mu M$ is given in Supplementary Fig. 3. Evaluation of compositional and thermodynamic parameters follows from the dilute phase response gradient $R_{AB}$ and the saturation concentrations. Source data are provided as a Source Data file.

within condensates. Indeed, we determined a FUS dominance of $D^{FUS} = 0.61 \pm 0.24$, meaning that only ~61% of the free energy decrease released by phase separation can be associated with FUS (Fig. 3e). This highlights the potential importance of the ion partitioning to the overall energetic driving force.

To investigate this further, we determined the fluorescence lifetime of the EGFP protein tag on FUS as a function of varying KCl concentrations. Here, a much shorter lifetime was observed in the dense phase (Supplementary Fig. 5), as driven by a higher refractive index due to the locally increased protein concentration within condensates[30,31]. The fluorescence lifetime of the condensed phase was determined to further decrease with decreasing KCl concentration (see Fig. 3e). Hence, FUS condensates appear to become more densely packed, indicative of enhanced protein interactions through the removal of potassium chloride ions. This further supports our observations above, that ion exclusion from the dense phase and potential

release from FUS polypeptide chains act as important contributors to driving phase separation.

Condensate assembly at low ionic strength is also commonly observed for other phase separation-prone proteins, which might suggest that the underlying partitioning behaviour with ions may be similar. To probe the flexibility and simplicity of our approach, we then sought to investigate ion partitioning and energetics for two further physiologically relevant phase separation-prone proteins in TDP43 and PGL3[32,33]. Upon determining the protein dilute phase concentrations under varying KCl concentrations by performing heterotypic line scan measurements for TDP43 and PGL3 (Fig. 3f, g, Supplementary Figs. 6 and 7 for time traces), we find condensate dissolution at increasing ion concentrations. Both TDP43 and PGL3 exhibited $K < 0$, indicating a similar ion exclusion mechanism to FUS. Despite the conserved ion exclusion behaviour, the response gradients, critical concentrations, and dominance values varied between the protein

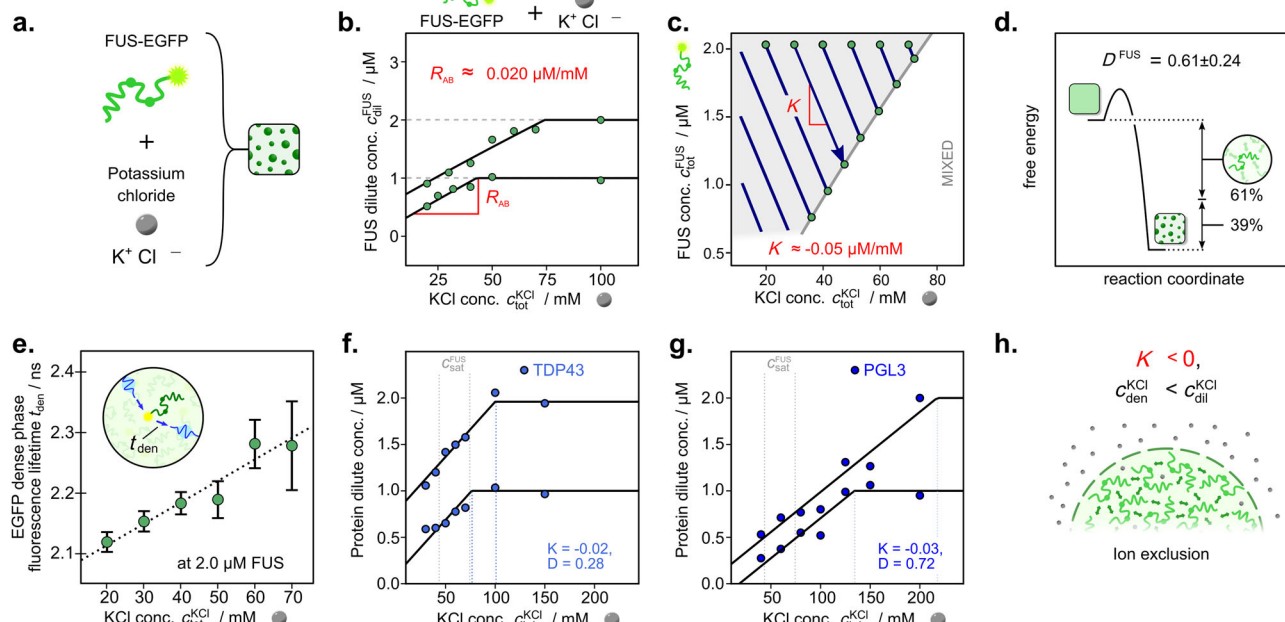

**Fig. 3 | Ion exclusion facilitates homotypic protein phase separation at low salt conditions. a** Phase separation of FUS-EGFP was studied in the presence of potassium chloride (KCl). **b** Dilute phase concentration changes of FUS at varying total KCl concentrations. **c** Determination of the FUS/KCl phase boundary from dilute phase concentration changes in (**a**). **d** Schematic representation of the dominance value of FUS. **e** Dense phase EGFP fluorescent lifetimes ($t_{den}$) at 2 μM FUS and varying KCl concentrations. Data are presented as mean ± SD from $n > 3$ repeats. Dilute phase concentration changes of proteins TDP43 (**f**) and PGL3 (**g**) at varying total KCl concentrations. Grey dotted lines highlight the FUS saturation concentration for comparison. **h** Schematic illustration of ion behaviour for FUS phase separation at low KCl concentrations. Source data are provided as a Source Data file.

systems, highlighting the impact of distinct sequence features. Here, PGL3, for example, presented the highest dominance fraction of 72%. This suggests an increased importance of protein-associated interactions, as corroborated by an increase in the critical concentration of KCl compared to FUS and TDP43. TDP43, on the other hand, displayed the highest relative ion partitioning, indicating that the KCl dense and dilute phase concentration difference is most significant in this system, therefore yielding the lowest dominance fraction of $D = 28\%$.

Together, we highlight the ability of this dilute phase concentration measurement approach to evaluate the collective interaction properties of protein condensates, revealing a conserved ion partitioning mechanism at low ionic strengths (Fig. 3h).

## Ions partition into condensates at high salt concentrations to drive non-ionic interactions

FUS, like many other proteins, commonly displays a so-called reentrant regime, where phase separation is observed at very high salt concentrations[34] (Fig. 4a). Prompted by the observation of differential ion partitioning at low ionic strengths, we set out to characterise collective interactions in the presence of high concentrations of lithium chloride (LiCl) and caesium chloride (CsCl). Both LiCl and CsCl have been previously shown to display large differences in the required saturation concentration[34]; however, the mechanistic basis of this behaviour has remained elusive.

We first characterised the change in dilute phase concentration of FUS-EGFP under varying concentrations of LiCl upwards of 4 M (Fig. 4b). A rapid drop-off in $c_{dil}^{FUS}$ at the onset of reentrant phase separation was observed at around 5 M LiCl (see Supplementary Fig. 8 for time traces), consistent with previous observations[34]. We then determined the ion partitioning of LiCl from the dilute phase concentration changes to find an enrichment of LiCl in condensates in this reentrant regime. The drastic increase in ionic strength drives non-ionic interactions such as π-stacking or non-polar contacts as well as hydrophobic interactions driven by interfacial water release[34]. This

likely causes the observed recruitment of ions into the condensed phase, which is expected to further strengthen these non-ionic interactions (Fig. 4c).

Next, we characterised changes in $c_{dil}^{FUS}$ as a function of CsCl concentrations (Supplementary Fig. 9) to elucidate the impact of changing the cation from Li$^+$ to Cs$^+$ (see Fig. 4d). Phase separation was already observed at approx. 3.8 M CsCl compared to a saturation concentration of around 5 M for LiCl. Therefore, Cs$^+$ is more potent at triggering reentrant phase separation of FUS, as also highlighted by the steeper drop-off in $c_{dil}^{FUS}$ with increasing CsCl concentration compared to LiCl (see Fig. 4b). CsCl further partitions less into FUS condensates, indicating that fewer ions are required in the dense phase to allow for sufficient non-ionic interactions to trigger phase separation. This is further highlighted by the fact that the tie line gradient appears to be closely linked to the solubility of the salts individually with $c_{sat}^{LiCl}$ being 13.4 M ($1/K = 143$ mM/μM) and $c_{sat}^{CsCl}$ being 5.9 M ($1/K = 64$ mM/μM) (according to supplier specification) as well as following the expected trend of the Hofmeister series[35].

The reentrant phase transition highlights extreme cases with regard to the energetics. At both tested protein concentrations (3 and 5 μM), the critical salt concentrations are largely similar. This yields a phase boundary gradient that approaches $P \to \infty$, with that $D^{FUS} \to 0$, meaning little free energy release is associated with the protein. This is because the system is entirely limited by the ion concentration, highlighting a critical stoichiometry that is necessary to trigger the formation of non-ionic interactions (Fig. 4e). In addition, $c_{dil}^{FUS}$ decreased to almost 0 μM protein concentration at high salt concentrations, suggesting that in this 'salting out' regime, the protein is almost completely sequestered into the condensed phase. Here, the phase boundary gradient approaches $P \to 0$ and with it $D^{FUS} \to 1$, suggesting that the system moves from an entirely salt-limited state to a protein-limited state (Fig. 4e). This can be rationalised by the fact that the salt solubility in the absence of protein is significantly larger.

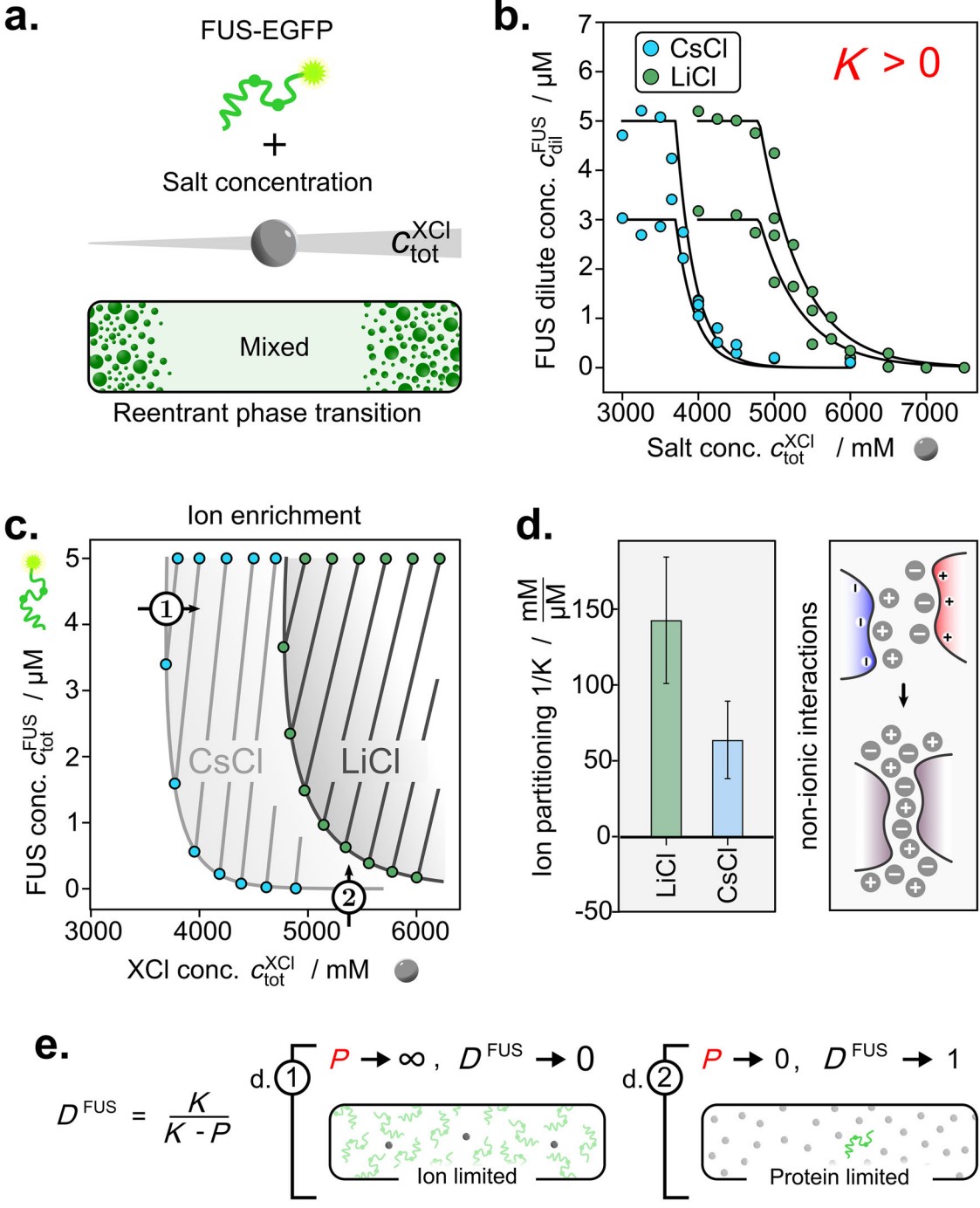

**Fig. 4 | Ions are enriched in FUS condensates under high salt reentrant conditions to foster non-ionic interactions. a** FUS displays reentrant phase separation at high salt concentrations. **b** FUS dilute phase line scans at varying total LiCl (green) and CsCl (blue) concentrations. **c** Phase boundaries show that phase separation occurs only above the respective critical salt and FUS concentrations. **d** Comparison of ion partition ($1/K$) of LiCl and CsCl into FUS condensates. Preferential partitioning at high ionic strength drives enhancement of non-ionic interactions. Data are presented as mean ± SD, parameter errors are estimated by repeated perturbation on fitting data and quantifying the spread of best-fit parameters. **e** High salt FUS phase separation indicates extreme cases where phase separation is driven by only the limiting component. This leads to dominance values of 0 or 1 depending on the position on the phase boundary. Source data are provided as a Source Data file.

Together, our experiments reveal that even simple ion species can display differential condensate partitioning, which can vary from enrichment to exclusion in a context-dependent manner.

## 1,6-hexanediol alters the solvation properties of biomolecules to destabilise condensates

1,6-hexanediol (1,6-HD) is commonly used as a generic inhibitor of hydrophobic interactions, and applying it has become a powerful strategy for studying phase separation. While 1,6-HD is used to investigate the nature of interactions underlying condensate formation or to assess the liquidity of condensates[36–39], it remains largely unclear how the protein physicochemical properties and collective interactions are altered as a result.

To shed light on the impact of 1,6-HD on collective interactions, we investigated FUS phase separation triggered by the addition of polyethylene glycol (PEG, 10 kDa) (Fig. 5a). We first characterised the

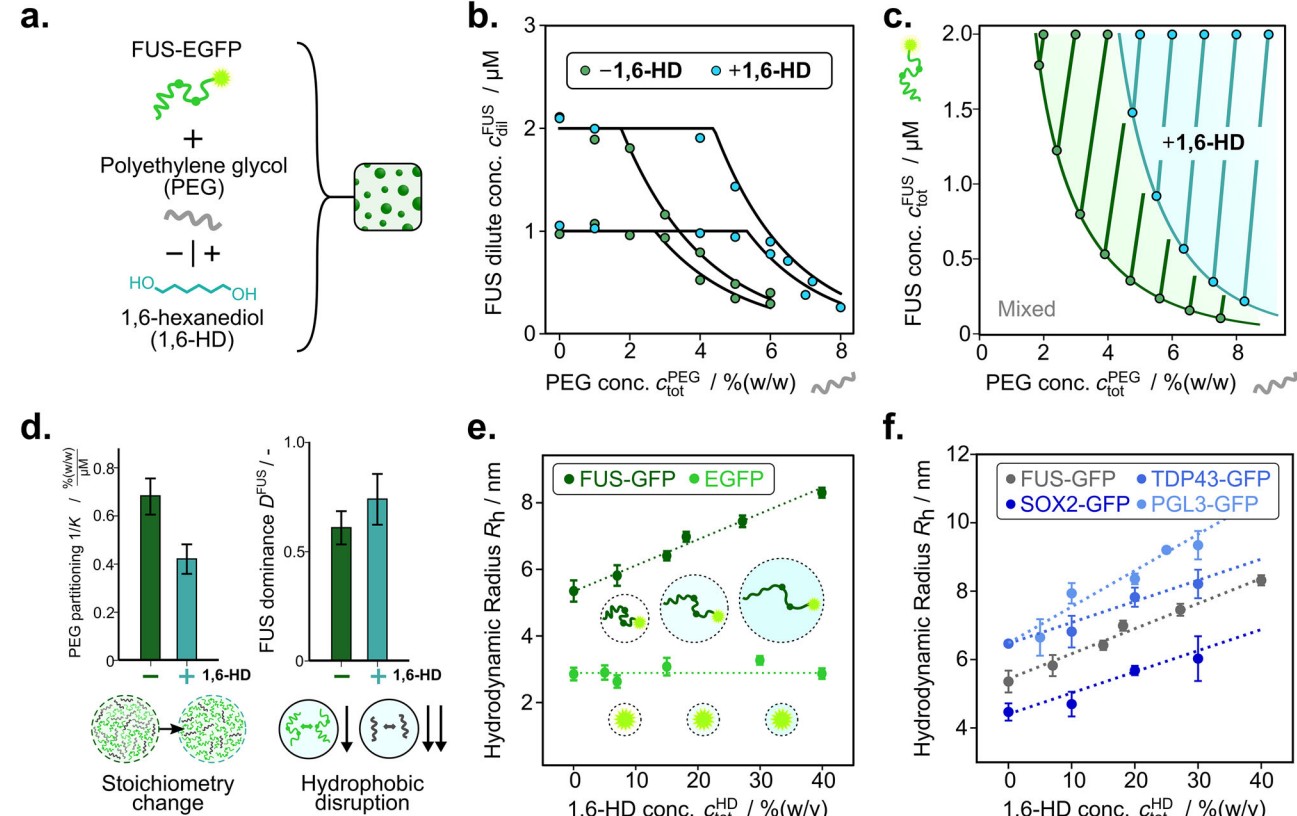

**Fig. 5 | 1,6-hexanediol (1,6-HD) decreases the phase separation propensity by modulating solvent quality. a** To characterise the mechanism of action of 1,6-hexanediol, phase separation of FUS with PEG was studied in the presence (+) and absence (−) of the compound. **b** FUS dilute phase concentration changes with varying concentrations of polyethylene glycol (PEG) in the presence (+, green) and absence (−, blue) of 1,6-HD (7% (w/w)). **c** Phase boundaries for FUS against PEG in the absence and presence of 1,6-HD were created from changes in dilute phase concentrations and tie line gradients. **d** (left) Shift in PEG partitioning (1/K) in the absence and presence of 1,6-HD, indicating a change in condensate stoichiometry by decreasing PEG/protein ratio. (right) FUS dominance in the presence and absence of 1,6-HD. Data are presented as mean ± SD, parameter errors are estimated by repeated perturbation on fitting data and quantifying the spread of best-fit parameters. **e** Changes in hydrodynamic radius $R_h$ as a function of 1,6-HD concentration. Shown are data for monomeric FUS-EGFP (dark green) and EGFP protein alone (light green) at 1 μM protein concentration each. Data are presented as mean ± SD from $n = 3$ repeats. **f** Change in hydrodynamic radius for a set of proteins with increasing 1,6-HD concentration. Data are presented as mean ± SD from $n = 3$ repeats. Source data are provided as a Source Data file.

system in the absence of 1,6-HD for which line scans were performed at a total FUS concentration of 1 and 2 μM where the total PEG concentration was varied from 0–8 % (w/v) (Fig. 5a, Supplementary Figs. 10 and 11). In the absence of 1,6-HD, phase separation of FUS was observed already at a PEG concentration of 2–3% (w/v), see Fig. 5b. A positive $K$ was determined between FUS and PEG, indicating that PEG co-partitions into condensates, rather than acting as a traditional crowder. This is in agreement with previous observations of PEG co-phase separating with protein condensates[18,40] and suggests potential favourable protein-PEG interactions.

Next, we repeated this experiment in the presence of 1,6-HD, shifting the phase boundary towards higher concentrations of PEG (Fig. 5c). The reduced tie line gradient between FUS and PEG was observed to decrease upon addition of 1,6-HD compared to the FUS and PEG system alone (Fig. 5d). From the partitioning behaviour, the dense-phase stoichiometry of constituent components can be inferred[18]. The results suggest that the presence of 1,6-HD decreases the number of PEG copolymers per FUS molecule by approximately half within condensates. Such a decrease in copolymer partitioning points towards a decrease in the interaction strength of PEG within the condensed phase. Dominance analysis shows that the free energy decrease associated with FUS does not change significantly, indicating a decrease in FUS self-interaction. This could be explained by 1,6-HD functioning as a solvation agent of the protein by decreasing the solvent polarity. The compositional changes can be traced back to the

fact that FUS can also engage in electrostatic interactions, while PEG, with only ether functional groups, will be less capable of doing so. FUS, however, is not capable of fully compensating for these effects with its electrostatic interactions, which leads to dissolution.

An increase in solvent interaction of the intrinsically disordered FUS protein would be expected to lead to an expansion of the polypeptide chain. Using microfluidic diffusional sizing[41], we determined the hydrodynamic radius $R_h$ of monomeric FUS-EGFP at different concentrations of 1,6-HD. We find a significant increase in the protein hydrodynamic radius with increasing 1,6-HD concentrations (Fig. 5f). The hydrodynamic radius of the folded EGFP protein alone, in comparison, displayed no change. This suggests that 1,6-HD acts as a solvation agent for the FUS disordered polypeptide chain that inhibits the intramolecular contacts driving protein compaction to favour solvent interactions, leading to an expansion of the protein size. Interestingly, this solvation effect is not limited to just the disordered protein. A similar expansion was also observed for the PEG copolymer (see Supplementary Fig. 12). This highlights that 1,6-HD effectively modulates the solvent quality for biomolecules to dissolve condensates.

We then set out to investigate if the observed 1,6-HD polypeptide chain expansion effect also translates to other protein systems. We tested for 1,6-HD mediated size expansion of GFP tagged variants of TDP43, PGL3 and SOX2, phase-separating proteins known to be sensitive to 1,6-HD dissolution[34]. In all cases, we observed an increase in the hydrodynamic radius upon addition of 1,6-HD (Fig. 5f), suggesting

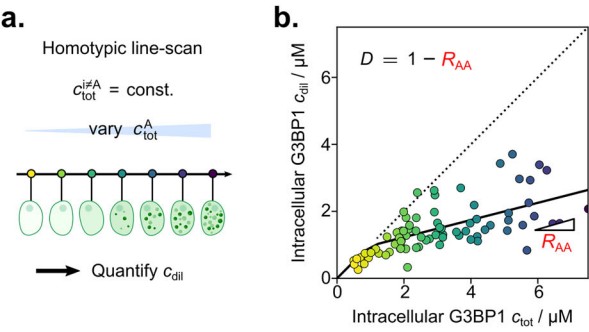

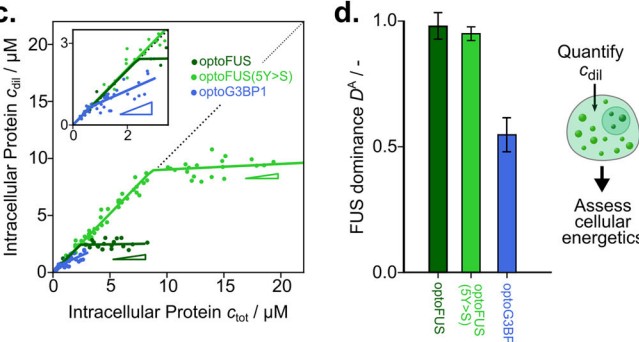

**Fig. 6 | Intracellular dominance determination. a** Homotypic line scans are performed by quantifying dilute phase concentrations of component A while varying the total concentration of component A (illustrated by yellow-purple colour gradient). **b** Homotypic response function of G3BP1 *in cellulo* from ref. 42. **c** Comparison of homotypic response functions of opto-protein (opto-FUS …

green, opto-FUS mutant 5Y > S … light green, opto-G3BP1 … blue) fusion constructs intracellularly from refs. 42,43. **d** Dominance fraction comparison of different proteins in (**c**). Data are presented as mean ± SD, parameter errors are estimated by repeated perturbation on fitting data and quantifying the spread of best-fit parameters. Source data are provided as a Source Data file.

that it more generally acts as a condensate dissolver by regulating the solvent-IDR interactions. We further sought to contrast the behaviour of these homotypic phase separation-prone proteins[34] with G3BP1, which is known for its RNA-mediated condensation[12]. G3BP1 did not display such polypeptide chain expansion with increasing 1,6-HD concentration (see Supplementary Fig. 13). This is expected as G3BP1, in the absence of RNA, adopts a self-inhibiting, compacted conformation driven by electrostatic interactions between oppositely charged intrinsically disordered regions (IDRs)[12].

Overall, our data highlights that measuring dilute phase concentrations only can enable the precise evaluation of the mechanism of action of condensate modulators. In the case of 1,6-hexanediol, this reveals a hydrophobic disruption effect driven by changes to the effective protein-solvent interaction.

### Application to complex environments

Lastly, we set out to translate our approach for the quantification of collective interactions to complex environments, such as *in cellulo*. Specifically, we can amend our dilute phase response function measurement approach to focus only on the protein[19] of interest by performing so-called homotypic line scans. Here, the concentration of the protein (component A) is varied while the concentrations of the other components i ≠ A are kept constant (Fig. 6a). This allows us to quantify the energetics of phase separation without having to vary the concentration of an additional species, such as salt or copolymers. By plotting the total protein concentration against the dilute phase protein concentration, the homotypic response gradient ($R_{AA}$) can be determined. At concentrations below $c_{sat}$, the dilute phase protein concentration increases linearly with the total protein concentration with $R_{AA} = 1$. Above $c_{sat}$, the protein is recruited into the dense phase, giving $R_{AA} < 1$, where the dominance can be determined as (see "Derivation of the dominance framework" section):

$$D_A = 1 - R_{AA} \qquad (4)$$

We then reanalysed previously published data from Riback et al.[42] where the intracellular dilute phase concentration post phase separation was quantified for G3BP1 at increasing expression levels (Fig. 6b). This shows a response gradient $R_{AA}$ of 0.26 ± 0.04, indicating that when observing increased G3BP1 concentrations, the dilute phase also rises. Hence, the added G3BP1 molecules are not just incorporated into the dense phase. This gives a free energy dominance fraction of ~74% indicating that other components also contribute to the free energy decrease.

We then turned to previously published data on opto-protein systems involving FUS and a tyrosine/serine mutation variant of FUS (5Y > S) as well as G3BP1[42,43]. In these opto-systems, as described by Riback et al.[42] and Wei et al.[43], the protein of interest is fused with Cry2, which can oligomerize in a light-inducible manner to trigger phase separation. To evaluate the intracellular dominance, we again plotted the dilute phase protein concentration against the total protein concentration for all protein systems (Fig. 6c). From the resulting response gradient, we find that opto-FUS displays a dominance of approximately one. Hence, despite the increased complexity *in cellulo*, opto-FUS is essentially the sole driver of phase separation. The introduction of the additional interaction mode through Cry2 appears to modulate the overall collective interactions to cause opto-FUS to behave effectively as a one-component system.

We then set out to quantify the effect of mutational changes to the protein sequence by performing dominance quantification for the 5Y > S mutant. Opto-FUS 5Y > S displays an increase in the protein $c_{sat}$ as expected from the removal of the interaction-prone tyrosine residues. The dominance fraction of FUS 5Y > S, however, remained approximately equal to one. Hence, while higher protein concentrations are required to yield phase separation for the 5Y > S mutant, the energetic driving force still purely stems from homotypic opto-FUS interactions. This is in stark contrast with opto-G3BP1, where a sloped response gradient is observed. This indicates that the opto-G3BP1 system displays a multicomponent character, as a fraction of the excess protein is not directly converted to the dense phase. Despite the addition of the Cry2 motif, G3BP1 therefore, is still reliant on heterotypic interactions with, e.g., RNA.

As such, using our measurement approach, protein-dependent energetics of condensation can be assessed directly *in cellulo* by determination of the dilute phase concentration of a single component only.

## Discussion

Here, we have formulated a generic approach for the quantification of collective interactions in biomolecular phase separation. Our strategy captures both compositional and energetic details of biomolecular phase separation and requires measurement of dilute phase concentrations of a single component only (Fig. 7a). We have discovered that protein phase separation at low ionic strength conditions commonly drives preferential exclusion of ions from condensates to decrease charge screening (Fig. 7b). At very high salt concentrations an opposing trend is observed, where ions preferentially partition into condensates to enable non-ionic interactions (Fig. 7c). Our findings also demonstrate that 1,6-HD disrupts condensates by functioning as a solvation agent that drives polypeptide chain expansion (Fig. 7d).

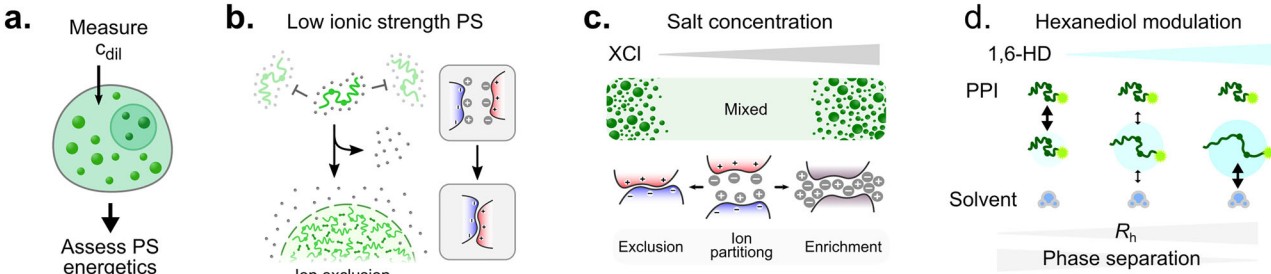

**Fig. 7 | Quantification of collective interactions enables deciphering the physicochemical driving forces of phase separation. a** Dilute phase concentration measurements of one component only can enable assessment of the energetics of phase separation processes. **b** For phase separation triggered by a decrease in ionic strength, an ion exclusion mechanism from the dense phase was observed across multiple proteins. **c** Ion partitioning displays opposing behaviours in phase separation at high and low ionic strength conditions. **d** 1,6-HD dissolution effects were observed to correlate with an increase in polypeptide chain expansion.

Our investigations of protein phase behaviour at different ionic strength regimes highlight that even simple molecules can display distinct partitioning and stabilisation behaviour. Specifically, in the low salt regime, KCl acts to prevent phase separation as it inhibits protein interactions by charge screening. Therefore, phase separation occurs upon decreasing the salt concentration, and it becomes favourable to exclude ions from the dense phase. Conversely, in the high salt regime, the increase of LiCl or CsCl salt concentration acts as the trigger for phase separation by driving non-ionic interactions, favouring ion enrichment. More generally, this might indicate that the partitioning of molecular species is largely a consequence of their propensity to enhance or counteract collective interactions.

1,6-Hexanediol is widely used as a modulator of protein phase separation to probe physicochemical driving forces, dissolution effects, or material properties[34,36–39]. In this context, 1,6-HD is generally recognised as a disruptor of hydrophobic interaction[34]. Our analysis reveals that this effect stems from 1,6-HD acting as a solvating agent for biomolecules. Specifically, 1,6-HD decreases intra- and intermolecular interactions by favouring solvent interactions, as corroborated by an expansion of the polypeptide chain. Our mechanistic study of 1,6-HD suggests that affecting the biomolecule's native state and chain expansion is a powerful avenue for modulating phase behaviour. Hence, screening for molecules capable of inducing expansion or compaction of intrinsically disordered proteins might be a simple but effective drug discovery strategy.

The presented approach allows for the study of the behaviour of species, such as salt ions or small-molecule compounds, which would otherwise be inaccessible by conventional labelling or detection strategies. This makes the assay especially suited to applications in more complex systems and in cellulo, as the characterisation of the underlying driving forces can be performed by measuring the response of a single component only. The dilute phase centred strategy presents a number of important advantages, such as drastically decreasing material requirements, and dilute phase readouts can also be generated easily using commonly available measurement assays.

Taken together, the quantification of collective interactions with the presented approach promises to be a powerful tool for studying phase separation in a wide range of contexts. We envision that the approach could be applied broadly to gain mechanistic insights to rationally design condensate modulators; dissect the impact of potential drug candidates; study the effect of biologically relevant molecular species on in vivo phase transitions; and optimise designer phase separation systems for functional applications.

## Methods

### Fabrication of microfluidic devices
Microfluidic devices were designed using AutoCAD software (Autodesk), followed by printing on acetate transparency masks (Micro Lithography Services). The replica master was obtained via standard soft-lithography steps using spin-coating of SU-8 photoresists (MicroChem) onto polished silicon wafers[44]. Typically, SU-8 3050 was applied to achieve a device height of approximately 100 μm. After UV exposure, using a custom-built LED-based apparatus[45], the precise heights of the features were measured using a profilometer (Dektak, Bruker). Devices were then produced in polydimethylsiloxane (PDMS). PDMS (Dow Corning) was mixed in a 10:1 (w/w) ratio with curing agent (Sylgard 184, Dow Corning) and poured onto the master, followed by degassing and baking for 1.5 h at 65 °C. The PDMS was then removed from the master and punched using biopsy punches to generate inlet holes, after which the slab was bonded onto thin glass slides using oxygen plasma surface activation (Diener electronic, 40% power for 30 s).

### Reagents
Potassium chloride (KCl), Lithium Chloride (LiCl), Caesium Chloride, polyethylene glycol (PEG, 10 kDa), Tween 20, Tris(hydroxymethyl) aminomethane (TRIS), and 1,6-hexanediol (1,6-HD) were purchased from Sigma–Aldrich (United Kingdom).

### Protein expression
EGFP-tagged fused in sarcoma (FUS) was expressed as reported previously[13] in an insect cell expression system[13]. After purification, the protein was stored at −80 °C in 50 mM TRIS buffer (pH 7.4), 500 mM KCl and at a total protein concentration of 70 μM. PGL3-6xHis-mEGFP was obtained from a previously published purification protocol[32]. The protein was stored at −80 °C followed by aliquoting and flash freezing in liquid nitrogen. EGFP-tagged TDP43 was expressed as described previously[34] from Sf9 insect cells using a baculovirus system. The purified protein was then stored at −80 °C in TRIS buffer (pH 7.4, 50 mM), 500 mM KCl, 5% (w/v) glycerol and 1 mM DTT. Similarly, His$_6$-MBP-Sox2-GFP was also expressed in Sf9 insect cells. Following purification, the protein was stored at −80 °C in Bis-Tris-Propane (pH 7.5, 50 mM), as well as 500 mM KCl, 1 mM DTT and 5% glycerol. G3BP1-6xHis-mEGFP was produced using a previously published protocol[46] and post-purification stored in TRIS buffer (pH 7.4, 50 mM) including 300 mM KCl, 1 mM DTT and 5% glycerol.

### EGFP synthesis and purification
pET28a vector harbouring the gene encoding His-tagged enhanced green fluorescent protein (EGFP) was purchased from GenScript. 40 mL of lysogeny broth (LB) medium supplemented with 36 μg/mL of kanamycin was inoculated with one colony of E. coli BL21 DE3 cells (New England Biolabs) harbouring the vector encoding EGFP. The cells were grown overnight at 37 °C and 180 rpm. The next day, 40 mL of fresh LB-kanamycin media was inoculated with 1 mL of overnight culture and grown for 2–3 h, until the optical density reached -0.5 optical units. Then, the cell culture was supplemented with 1 mM of isopropyl-beta-D-thiogalactopyranoside (IPTG, Sigma–Aldrich) and grown for an additional 3 h. The cells were harvested by centrifugation at 5000 × g and

4 °C for 10 min. Harvested cells were resuspended in lysis buffer (1 mg/mL lysozyme, 10 mM imidazole in PBS (pH 7.4)) and incubated on ice for 30 min. Subsequently, cells were sonicated on ice using $6 \times 10$ s on/pause cycles (at 42% power cycle). Sonicated cells were centrifuged for 30 min at $10.000 \times g$ and 4 °C. The supernatant was collected and loaded onto 1 mL of equilibrated Ni-NTA slurry in a BioRad gravity flow column. The column was placed on a rocker and incubated in a cold room at 4 °C for 60 min. Subsequently, the column was placed in a ring stand. After the beads settled, the flow-through was allowed to run, then the column was washed with $4 \times 1$ mL of wash buffer solution (20 mM imidazole in PBS, pH 7.4). Subsequently, EGFP was eluted using 2 mL of elution buffer (250 mM imidazole in PBS, pH 7.4). The collected EGFP solution was buffer exchanged to 50 mM TRIS buffer (pH 7.4), 150 mM KCl using 10 kDa cut-off centrifugal filters (Millipore).

### Sample preparation and phase separation conditions

Phase separation was induced by gently mixing the individual components in Eppendorf tubes. Briefly, in a first step, buffer was added to the Eppendorf tube. The buffer was 50 mM TRIS (pH 7.4) supplemented with 0.01% (v/v) Tween 20 in all cases. For FUS/KCl experiments, KCl stock solution (150 mM KCl in 50 mM TRIS (pH 7.4)) was then added to achieve the desired total KCl concentration, also considering the amount of KCl introduced from the protein stock. This was followed by the addition of FUS from the protein stock (70 μM FUS-GFP in 50 mM TRIS buffer (pH 7.4), 500 mM KCl). For the FUS/KGlu experiments, the KCl concentration was kept constant at 40 mM, also considering the amount from protein addition later on. Then, KGlu stock solution (1000 mM KGlu in 50 mM TRIS, pH 7.4) was added to achieve the desired KGlu concentration prior to the addition of FUS stock solution. For FUS/1,6-hexanediol and GUG aptamer experiments, 150 mM KCl and 15% (w/v) PEG (in 50 mM TRIS (pH 7.4), 150 mM KCl) were added to the buffer. Then, either 1,6-hexanediol (50% (w/v) in 50 mM TRIS (pH 7.4), 150 mM KCl) or GUG aptamer (100 μM in 50 mM TRIS (pH 7.4), 150 mM KCl) were added, followed by the addition of pre-diluted FUS stock (5 μM FUS-GFP in 50 mM TRIS buffer (pH 7.4), 150 mM KCl). This yielded solutions containing 5% (w/v) PEG with 7% (w/v) 1,6-hexanediol in 50 mM TRIS buffer (pH 7.4), 150 mM KCl in the case of the 1,6-hexanediol experiment series or varying GUG concentrations between 1 and 1000 nM in GUG aptamer experiments. In all cases, typically 10 μL of sample was prepared and final FUS concentrations were 1 or 2 μM.

### Dilute phase concentration measurements using confocal detection

Samples were prepared as described above by mixing first off-chip and incubating for 5 min prior to the experiment. Microfluidic chips were filled with sample buffer followed by flushing the sample for 5 min at 100 μL/h to ensure equilibration of the channel (see Supplementary Fig. 12). To operate flow control through the microfluidic channel, negative pressure created by a glass syringe (Hamilton, Switzerland) and syringe pump (neMESYS, Cetoni, Germany) was applied. Thereby, channel loading was controlled by an inlet reservoir containing buffer or sample that was connected to the channel inlet. Following equilibration, dilute phase concentrations were measured using a home-built confocal setup. Differences between devices utilised were minimal compared to the baseline noise level (see Supplementary Fig. 13). The setup is equipped with picosecond-pulsed 485-nm and 640-nm laser sources, a 60× water-immersion objective (CFI Plan Apochromat WI 60×, NA 1.2, Nikon), and an automated sample stage onto which the microfluidic device can be mounted. An illustration of the setup can be found in Supplementary Fig. 2. Following laser excitation of the sample through the objective in a diffraction-limited spot and collection of fluorescence through the same objective, emitted photons were registered via avalanche photodiodes and recorded using a time-correlated single-photon counting unit (HydraHarp400, Picoquant).

Laser synchronisation was done via a laser controller (Sepia PDL 828, Picoquant). The setup is further equipped with a lens-pinhole-lens system to remove out-of-focus light and with appropriate dichroic and bandpass filter sets to spectrally separate photons into blue- and red-emitting channels. The two lasers were run in pulsed-interleaved excitation (PIE) mode, driven by a synchronisation signal from the laser driver at a frequency of 25 MHz. Recording of data was done in T3 mode. Recorded macroscopic times $T$ were binned into 1-ms time intervals to give an intensity readout in units of number of photons per second. Signal collection from samples was typically performed for 30 s with collection times as low as 0.1 s being sufficient to calculate tie line gradients (see Supplementary Fig. 14). The intensity time traces generated in this manner typically display a stable baseline (see Fig. 2b), which corresponds to the dilute phase concentrations, as the volume fraction of the dense phase can be assumed to be significantly smaller than the dilute phase. The baseline value can then be extracted from the maximum in the intensity histogram (see Fig. 2c). The standard deviation of the baseline signal can be determined from numerical fitting to the lower intensity branch of the histogram. The higher intensity branch will show condensed species, as condensates exhibit high protein concentrations. Concentration conversion of the baseline intensities was performed by calibration via quadratic fitting to the intensity recorded of samples at different, known FUS concentrations under non-phase-separated conditions.

### Fluorescence lifetime measurements

Fluorescence lifetime decays were calculated by generating a decay time histogram from a large number of photons. This was achieved by calculating the time difference between the APD signal and the last synchronisation event for each recorded photon. To estimate the lifetime of GFP in the dense phase, photons that correspond to the GFP fluorophore in the condensates are selected by applying a high-pass filter to the intensity readout previously computed by binning the macroscopic time $T$ in 1 ms intervals. More specifically, photons with a macroscopic time $T$ that fall in the high-intensity time window are identified as signal from condensates, and the $t$ values of all these photons are binned to give a single fluorescent decay histogram. To extract the lifetime, we truncated the histogram away from the instrument response function (IRF) and fitted an exponential decay function, whereas the inverse of the decay constant corresponds to the lifetime.

### Microfluidic diffusional sizing

Microfluidic diffusional sizing was performed as described elsewhere[41]. Briefly, the Fluidity One-M instrument (Fluidic Analytics) was utilised for measurements. Auxiliary channels were first primed with 50 mM TRIS buffer (pH 7.4), 150 mM KCl. Subsequently, 3.5 μL of sample with 1 μM protein concentration and specified concentrations of 1,6-hexanediol were added. Each sample was measured three times. Detection was set for Alexa488 fluorescence detection, and size-range settings of $R_H = 2–9$ nm were applied on the Fluidity One-M instrument.

### Derivation of the dominance framework

We assign $c_{tot}^i$ as the total concentration of solutes i = A, B,…, N, while dense and dilute phase concentrations are given as $c_{den}^i$ and $c_{dil}^i$, respectively. The phase boundary is then given as a dense and dilute (N-1)-dimensional surface provided by the set of concentrations of $c_{den}^i$ and $c_{dil}^i$. Mass balance demands conservation of solutes, leading to the definition of the tie line vector:

$$c_{tot}^i = c_{dil}^i + v \times k^i$$

where $v$ is the dense phase volume fraction and $k^i = c_{den}^i - c_{dil}^i$.

Assuming a binary phase separation system, a fixed volume of particles and considering only small dense volume fractions, i.e. probing in close proximity to the dilute phase branch of the phase

boundary, we can compute the energy difference between the homogeneous and phase-separated state $\Delta f$ as[19]:

$$\Delta f = -\frac{1}{2}\sum_{i=1}^{N}\sum_{j=1}^{N} v \times k^i v \times k^j \partial_j \mu_i(c_{dil}) + O(v^3)$$

Here, $\mu_i$ is the chemical potential as $\mu_i(c_{tot})$ and as can be inferred from the definition of the tie line gradient $v \times k^i$ as the distance between the total and dilute phase composition. To access the solute species-specific difference $\Delta f^i$ we consider the sum over $j$ to find:

$$\Delta f^i \equiv -\frac{1}{2} v \times k^i \sum_{j=1}^{N} v \times k^j \partial_j \mu_i(c_{dil})$$

At the onset of phase separation, i.e., $v \to 0$, this becomes:

$$\lim_{v \to 0} \frac{\Delta f^i}{\Delta f} = \frac{\sum_{j=1}^{N} k^i k^j (\partial_i \partial_j f)|_{c_{dil}}}{\sum_{l=1}^{N}\sum_{n=1}^{N} k^l k^n (\partial_l \partial_n f)|_{c_{dil}}} \equiv D^i$$

yielding the dominance of species i. To connect this to accessible quantities, we first introduce the dilute phase boundary normal vector: $\sum_{i=1}^{N} n_i \delta c_{dil}^i = 0$, with $\delta c_{dil}^i$ a vector on the dilute phase boundary branch. Applying equilibrium perturbations to obtain $n_i \propto \sum_{j=1}^{N} k^j (\partial_i \partial_j f)|_{c_{dil}}$[19], we can rewrite the dominance as:

$$D^i = \frac{n_i k^i}{\sum_{l=1}^{N} n_l k^l}$$

Further, quantification of all species concentrations is typically inaccessible. This leads us to focus on the quantification of the dilute phase concentration of a selected species, which can be quantified experimentally using simple separation of the dense phase from the dilute phase or application of confocal techniques[18,42]. Hence, we focus on the variation of the dilute phase concentration of an individual species i, upon applying perturbations of the total concentration of another species j to define the response function

$$R_{ij} \equiv \frac{\partial c_{dil}^i}{\partial c_{tot}^j}$$

which can be rewritten by applying mass balance to give[19]:

$$\lim_{v \to 0} R_{ij} = \delta_{ij} - \frac{n_j}{n_i} D^i$$

with $\delta_{ij}$ the Kronecker-Delta. Now, in a specific example involving the primary species A and B, we can then further look to identify a set of dilute phase concentrations of component A that do not change $(\delta c_{dil}^A = 0)$ while the total concentration of another solute species B is varied. Furthermore, we consider that the concentration of all other species is kept constant, i.e., $c_{tot}^j = 0$ for j other than A or B. Considering the definition of the response function, this becomes $\delta c_{dil}^A = R_{AA} \times \delta c_{tot}^A + R_{AB} \times \delta c_{tot}^B$ since all other components $\delta c_{tot}^j = 0$ for $j \neq A,B$. This allows us to introduce the 2-D reduced tie line gradient:

$$K \equiv \left(\frac{\partial c_{tot}^A}{\partial c_{tot}^B}\right)_{c_{dil}^A} = -\frac{R_{AB}}{R_{AA}}$$

which describes the line defined by $\delta c_{dil}^A = 0$ in the A-B plane. Again, considering only investigations at the onset of phase separation, we can use the previous relation between $R_{ij}$ and $D^i$ to give:

$$\lim_{v \to 0} K = \frac{n_B}{n_A} \frac{D^A}{1 - D^A}$$

Now defining P as the phase boundary gradient vector in the A-B plane, which is orthogonal to the normal vector $\left(P = -\frac{n_B}{n_A}\right)$ and staying close to the phase boundary, we obtain maintext Eq. 1:

$$D^A = \frac{\Delta f^A}{\Delta f} = \frac{K}{K - P}$$

## Experimental characterisation of response and phase boundary gradient

The dilute phase response gradient $R_{AB}$ is determined directly from dilute phase line scan data. Line scans correspond to the change in dilute phase concentration of component A ($c_{dil}^A$) at constant total concentration of component A ($c_{tot}^A$ = const.) but varying total concentration of component B (vary $c_{tot}^B$). Here, the concentration of component B is varied from outside the two-phase coexistence region to well within the phase-separated regime. This is repeated at two separate total concentrations of component A ($c_{tot,1}^A$, $c_{tot,2}^A$) to give two line scans. Both line scans are then individually fitted in the experimentally probed concentration range to give a function $c_{dil}^A = f(c_{tot}^B)$. In the case of FUS and KCl (A = FUS, B = KCl), for example, the data shows a linear response to changes in KCl; therefore, the following form was chosen for phenomenological fitting to both line scans at $c_{tot,1}^A$ and $c_{tot,2}^A$.

$$c_{dil}^A = \begin{cases} c_{tot}^A, & c_{tot}^B > c_{sat}^B \\ a + b*c_{tot}^B, & c_{tot}^B < c_{sat}^B \end{cases}$$

In doing so, a, b and $c_{sat}^B$ can be determined for both line scans. For this particular form of the dilute phase response, the response gradient $R_{AB}$ is simply given as the slope of the line scan fits at the phase boundary (e.g., $R_{AB} = (b_1 + b_2)/2$, if the slope changes between line scans). The dilute phase boundary gradient P is then determined as follows:

$$P = \frac{c_{tot,2}^A - c_{tot,1}^A}{c_{sat,2}^B - c_{sat,1}^B} = \frac{\Delta c_{tot}^A}{\Delta c_{sat}^B}$$

With R and P in hand, the reduced tie line gradient K, as well as the component A dominance $D^A$ is calculated:

$$D^A = \frac{\Delta f^A}{\Delta f} = \frac{K}{K - P} \text{ and } K = \frac{R_{AB}*P}{R_{AB} - P}$$

## Phase diagram extrapolation

The phase boundary represents the set of conditions of dilute ($c_{dil}^A$, $c_{dil}^B$) and dense ($c_{den}^A$, $c_{den}^B$) phase concentrations formed as a result of the demixing of a set of total concentration conditions ($c_{tot}^A$, $c_{tot}^B$) within the two-phase coexistence region. Here, the dilute phase part of the phase boundary can be extrapolated from component A dilute phase response function data only, as reported previously[18]. Briefly, for a specific condition ($c_{tot,1}^A$, $c_{tot,1}^B$) the experimentally determined dilute phase line scans provide the corresponding dilute phase concentration $c_{dil,1}^A$. The dilute phase concentration of component B $c_{dil,1}^B$ can then be determined by following the tie line (with reduced tie line gradient K) originating in ($c_{tot,1}^A$, $c_{tot,1}^B$) until $c_{dil,1}^A$ (see Supplementary Fig. 4). Thereby, the point on the dilute phase branch of the phase boundary corresponding to the demixing of ($c_{tot,1}^A$, $c_{tot,1}^B$) can be simply written as:

$$(x, y) = (c_{dil,1}^A, c_{dil,1}^B) = (c_{dil,1}^A, c_{tot,1}^B - (c_{tot,1}^A - c_{dil,1}^A)/K)$$

By assuming a constant tie line gradient and employing previously mentioned phenomenological fitting for the line scans, this can be

repeated for any ($c_{tot}$, $n^A$, $c_{tot,n}^B$) within the experimentally characterised concentration range to determine the corresponding phase boundary condition ($c_{dil}$, $n^A$, $c_{dil,n}^B$).

## 2-D tie line reduction

A reduced tie line is by definition a collection of points on the 2-D experimental phase diagram that have the same dilute phase component A concentration (see Supplementary Fig. 1). When the full phase space is 2-D, the phase boundary and tie lines all lie on a plane, and near the dilute phase boundary points on each tie line all have the same dilute phase A concentration. As such, in the 2-D scenario, the reduced tie line constructed using component A dilute phase concentration is equivalent to the higher-dimensional tie line, and is also equivalent to the reduced tie line constructed using dilute phase B concentration. On the other hand, when the full phase space is 3-D, the phase boundary itself becomes a 2-D surface. The collection of points with the same component A dilute phase concentration constitutes another surface, since 3 (dimensions) − 1 (constraint) = 2 (degrees of freedom). This surface is a flat plane perpendicular to the component A concentration axis in the mixed region, where the dilute phase concentration is equivalent to the total concentration. When propagating this surface into the phase-separated region, one has to bear in mind that many tie lines in the 3-D space share the same dilute phase A concentration, and these tie lines can be identified by the intersection between the trivial flat plane at $c_{tot}^A$ = const. and the 2-D phase boundary. We identify the tie lines of interest as the ones with one end on this intersection curve, and stitching them together produces a 2-D surface that divides the phase-separating region into two parts. This newly generated surface can be thought of as a tie plane with respect to component A, and it is, in general, curved. When line scans are performed in a 2-D experimental cross-section of the full 3-D space, this cross-section intersects the tie plane, and this intersection is the reduced tie line measured here. Hence, the reduced tie line depends on both tie lines and the phase boundary itself when more than 2 components are present.

## Reporting summary

Further information on research design is available in the Nature Portfolio Reporting Summary linked to this article.

## Data availability

Source data are provided with this paper, including the raw data dilute phase concentration data in the associated Supplementary Figs. Source data are provided with this paper.

## Code availability

Exemplary analysis code for the determination of tie line gradients is provided with the source data. Additional analysis steps have been described in detail in the "Methods" section.

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

## Acknowledgements

The research leading to these results has received funding from Global Research Technologies, Novo Nordisk A/S (H.A., T.P.J.K.), the European Research Council under the European Union's Horizon 2020 Framework Programme through the Marie Sklodowska-Curie grant MicroSPARK (agreement no. 841466; G.K.), the Herchel Smith Funds (G.K.), the Alzheimer's Association Zenith Award (P.S.G.-H.) and the Wolfson College Junior Research Fellowship (G.K.), and the Frances and Augustus Newman Foundation (E.d.C., T.S.). T.J.W. thanks the Harding Distinguished Postgraduate Scholar Programme.

## Author contributions

H.A., D.Q., G.K., T.J.W. and T.P.J.K. designed and conceptualised the study. H.A. and E.d.C. performed experiments. G.K., T.J.W., T.S., T.M.F., S.Q., N.A.E., S.A., A.A.H., P.S.G.-H. and T.P.J.K. provided materials and methods. H.A. and D.Q. analysed the data. H.A., D.Q., E.d.C., G.K., T.J.W., J.N.A., M.K., R.V.P. and T.P.J.K. interpreted data. H.A., G.K. and T.P.J.K. wrote the original draft of the paper. All authors discussed the results, commented on the manuscript, and contributed to the final manuscript.

## Competing interests

The authors declare no competing interests.
