## [Transparent Peer Review file · Nature Communications]

Quantifying collective interactions in biomolecular phase separation

Corresponding Author: Professor Tuomas Knowles

Version 0:

Reviewer comments:

Reviewer #1

(Remarks to the Author)

In this manuscript, the authors have formulated a method for quantifying the molecular interactions involving all the components of a biomolecular condensate by measuring dilute or dense phase concentrations of only one component. By this, one can derive a quantitative and a collective information regarding all the components in the system such as stoichiometry, free energy of contribution, etc., by measuring phase boundaries and tie-lines of only one component (component A) and its dilute phase concentrations (C_{dil}^A) as a function of the second component. This method of analysis could be of very useful to estimate the boundary conditions of phase separation quantitatively to derive important characteristics of the condensate. Therefore, this manuscript can be highly impactful for the biomolecular condensate community. The authors may consider addressing the following to resolve or clarify a few issues that will help better the manuscript.

It would be helpful for the reader if the derivation of relationship (Eq 1) is elaborated in methods or supplementary section:
 $D^A = \left[\frac{\Delta f}{\Delta A} \right] \frac{A}{\Delta f} = K/(K-P)$ (Eq 1)

The explanation for FUS undergoing LLPS in low ionic strengths is sound (page 4; paragraph 2) and well supported by the formulation and the models. However, this has to protein sequence-specific and cannot be generalized.

The data in the manuscript soundly demonstrates the methodology works for a binary component system (FUS and salt). However, it would greatly help if the authors could demonstrate/comment on how these will be helpful for multi-component systems.

Can the authors comment on why the slopes of C_{dil}^{FUS} line scans at low salt concentrations are linear while they are non-linear for re-entrant phase separation at high ionic strengths? Can a second 'R' value be calculated for re-entrant conditions and what would it inform us?

In the determination of R, wouldn't it be better to average slopes of three line-scans than two?

Since reduced tie-lines within the demixed phase describe compositions along which the dilute phase concentrations of a component does not vary, it is unclear from Fig 3c, as to what is indicated as C_{tot} and C_{dil} along the tie-line.

Minor points

In figures 2-5, it would be good to indicate what the color codes of circular symbols mean in the legends. As is, it remains as an interpretable aspect.

Figures 4d,e & f are not cited in the text.

Author contributions for AAH, and SA are not mentioned.

Reviewer #2

(Remarks to the Author)

In this work, Ausserwöger et al. present a new strategy to quantify collective interactions in biomolecular phase separation.

By performing line scans on samples with a constant concentration of fluorescently labeled component A (FUS) and varying concentrations of component B (e.g. KCl), phase separation of component A into a dense phase can be quantified through detection of a spike in fluorescence. From this data, the phase boundary between components A/B is determined, and key parameters such as the tie line gradient K and phase boundary gradient P are extracted. Finally, the dominance D of component A can be determined using K and P , quantifying the relative contribution of A to the free energy decrease of phase separation. The authors suggest that intracellular chloride ions may modulate FUS condensates in vivo. However, they only present in vitro data, thus applicability in vivo remains unknown. Similarly, the insight into the behavior of 1,6-hexandiol as a solvating agent is interesting, but their conclusions are supported with observations of FUS interactions/condensates, so it is not clear if that observation is generalizable to other protein condensates. Overall, while this is an interesting and potentially useful method, it is unclear the findings from the FUS system in vitro are generalizable to other condensates and in vivo. Further, some of the conclusions seem overstated, given the lack of experimental evidence, I cannot recommend publication in Nature Communications.

1. The data shown on FUS and ion concentrations is consistent with the proposed mechanism (i.e. intracellular chloride ions may modulate FUS condensates in vivo), but the conclusions in general are overstated. The authors observe correlations with the ion additions, but have no evidence for the molecular mechanism behind their macroscopic observation. Furthermore, it is unclear how generalizable the observations for effects of ions and hexanediol on FUS phase behavior are to other proteins, with other surface charge densities and conformations. In lieu of this, the authors should tamp down their claims, or provide evidence for the proposed mechanism as well as evidence for the observed behavior for other proteins.
2. Lines 317-318 are overstatements that require more disclaimers or experimental evidence for this behavior in vivo. The causative agent for ion release has not yet been determined, and this in vivo relevance depends entirely on the ions driving phase separation behavior of FUS. There may be any number of in vivo mechanisms that turn this in vitro behavior on its head. For example, how would the cell pumping of ions against the concentration gradient affect this behavior?
3. The use of the phrase "collective interactions" should be reconsidered. While this method does report on the interactions between different components in a two-component phase separated system, it actually describes the macroscopic phase separation, so a better term would be "bulk interactions." The term "collective interactions" is confusing as authors do not show any emergent collective behavior, but only refer to pair-wise interactions.
4. Is there a reason why only low salt concentration with KCl and high salt concentration with LiCl and CsCl was studied, but no low concentration with LiCl? The varying effect on FUS was explained via the high ionic strength of LiCl/CsCl vs. KCl, so it would strengthen the claim by confirming this varied behavior at low salt concentrations of LiCl and CsCl.
5. It is mentioned that the dilute and dense phase concentrations are quantified using the intensity readouts from the linescans. How does this procedure work? The total concentration of protein is known, but how is the ratio of dilute to dense phase calculated based on the fluorescence readouts? Is photobleaching accounted for?
6. P. 2, line 77: The authors state that this approach could be combined with standard epifluorescence or absorbance-based measurements. How would standard epifluorescence work, given that it integrates the fluorescence signal in z ? Would it not only read out the total fluorescence of the dilute and dense phase, leading to the same result in every case?
7. The confocal microscope description shows two wavelengths used, both 485nm and 640nm together using Pulsed-Interleaved Excitation. The work describes using GFP tagged polymers, which explains the 485nm laser. I was not able to find what fluorophore is the 640nm laser exciting, and what is its purpose? Also, while it is implied from the supplementary figure, it should be stated explicitly whether this is a scanning or spinning disk confocal microscope in the Methods section.
8. Is there mixing of the prepared solutions upon loading into the channel? The SI states "following equilibration, dilute phase concentration were measured..." but this should be expanded to explicitly consider mixing. Is there a control for avoiding heterogeneity in the system due to bad mixing, as opposed to phase separation?
9. The hydrodynamic radius of FUS-EGFP measured for the hexanediol experiments is helpful. To support authors' claims about the behavior of ions, the effect of ions on the hydrodynamic radius of FUS-EGFP should also be quantified. Overall, the interpretation about the effect of ions is a bit of a stretch, given the experimental evidence presented. While the interpretation is feasible, the authors should provide an orthogonal measurement allowing to assess the changes in intra-FUS distance, hydrodynamic radius, etc in the presence of salts.
10. Lines 173-175, the observation that ion release triggers phase separation is overstated. A correlation of these two states was observed, but causation was not determined. The sentence should be reworded, or additional direct measurements should be reported to support this claim.
11. Line 190, strong non-ionic interactions arising from an increase in ionic strength is mentioned. Could authors list, which non-ionic interactions they are considering here?
12. Line 326, the authors mention "a multitude of commonly available measurement assays", but when introducing their method, they only list epi-fluorescence and absorbance. Are there any more? Please list. Alternatively, remove "multitude" as it evokes a high number of possible approaches.
13. The presented framework relies on a classic equilibrium thermodynamics. Have the authors considered entropy and enthalpy connected with such phase separation and its possible effect in vivo?
14. The applicability of these observations to in vivo systems is unclear, due to many obvious differences between the systems. Beside activity, etc, liquid condensates in vivo contain RNA as ubiquitous ingredient, with RNA being charged, flexible, etc, hence very likely influencing the effect of salts on proteins in condensates in vivo. Thus, it is difficult to draw conclusions about phase behavior of liquid condensates in vivo.

Minor Points:

1. Line 25 typo: "would enable to address"
2. Figure 4e and 4f are discussed but not explicitly referenced in the text anywhere.
3. Figure S3, subplots are not labelled, and referred to as "a", "b", and "left panel".
4. Supplementary Figure S2 is labeled as S1, and all subsequent SI figure titles are off by 1.

5. Figure 4a: “reentrant phase transition”

6. Figure 4c: “no-ionic interactions” should be “non-ionic interactions.”

Reviewer #3

(Remarks to the Author)

In this manuscript the author develop an experimental and theoretical framework in an attempt to better quantify interactions among polypeptides and small molecules that shape the landscape of biomolecular condensation. Their strategy is to understand liquid-liquid phase separation (LLPS) by characterizing the behavior of tie lines and component dominance, or fractional free energy decrease from one component relative to the overall free energy decrease from phase separation. To do so they utilize a microfluidic system to vary the concentration of a model disordered scaffold protein, FUS, along with salts and hydrophobic molecules, and measure the concentration of the FUS in the dilute phase using fluorescence confocal microscopy. By varying salt, KCl, concentration at two different FUS-EGFP concentrations they are able to determine saturation concentrations and tie lines to calculate the parameters P , K which can ultimately be used to calculate fractional dominance DA for the scaffold or one of other components. Using such a setup they first determine that the dominance of FUS-EGFP is 0.61, meaning $\sim 61\%$ of the total free energy decrease from LLPS can be attributed to FUS. The other portion is due to complex interaction with ions and water. Next, they characterize the effect on FUS density and fluorescence lifetime in the condensed phase as a function of low KCl concentrations. As previously known in the field, increased $[KCl]$ blocks FUS condensation, with a re-entrant LLPS phenomenon at very high concentrations. By mapping the dilute phase concentration of FUS relative to $[KCl]$ they determine a reduced negative tie line gradient ($K < 0$) indicative that the presence of KCl blocks FUS condensation. This trend indicates that KCL tends to be excluded from FUS condensates. Additionally, measurements of FUS-EGFP lifetime in dense phase that decreases at lower $[KCl]$, suggesting that dense phase becomes more packed at lower salt. Intriguingly at very high salt concentrations (5M LiCl), FUS displays re-entrant phase separation driven by non-ionic interactions. The authors also are able to extrapolate that LiCl enriches in FUS condensates under these conditions at a ratio of ~ 140 mM ion per micromolar of FUS. Finally, they explore the extent to which 1,6 hexanediol (1,6-HD) impacts FUS LLPS in the presence of PEG 10000, and FUS dominance. They propose that addition of 1,6-HD raises $csat$, lowers PEG partitioning and increases fractional dominance.

Overall, this is a clear and quantitative study of the parameters that contribute to collection interactions that drive disordered protein phase separation, authored by a number of leaders in the field. Strengths of the work include careful measurements of dilute phase FUS concentrations as a function of ions and other small molecules, calculation of tie lines and $csat$ s to determine the dominance of FUS, and illustrative diagrams and well written text to explain the trends in phase diagrams. Enthusiasm for the work is somewhat dimmed by the limited experimental nature of the study; the authors limit their characterization to well-defined phenomena. I have three major suggestions for improving the manuscript below

1. Significance or meaning of changes to fractional dominance of FUS. The authors calculate dominance of FUS as between $\sim 0.6-0.7$, dependent on the presence of KCl, PEG, 1,6-HD. However, it would be very helpful to additional examples in which fractional dominance shifts more significantly. What about use of a FUS mutant that has a higher $csat$ (such as from mutation of Arg or Tyr) or lower $csat$ (from addition of Tyr). Would this be expect to alter the dominance fraction? If it alters the hydrodynamic radius this in fact may be the case. Additionally mutation of Arg would affect interactions with ions. Alternatively, the authors should perturb the setup in some other fashion: altering pH would affect ionizable groups and interactions with KCl; they might test other types of salts. Framed another way, is there a theoretical reason to expect for a disordered protein with a $csat$ of ~ 2 μM that its dominance will never drop much below 0.5-0.6. Can they provide a simulation to support this. Alternatively, for context, what is dominance in a polyelectrolyte coacervate system: of polyanion and polycation; in this case is the fractional dominance of each ~ 0.35 and the ions also 0.3-0.4? If so, can the authors test a split version of FUS, into N-terminal FUS LC and C-terminal fragment that some of the authors previously demonstrated forms complex coacervates (Wang and Hyman, Cell 2018). In short, the manuscript would be greatly strengthened by provided additional cases in which dominance of FUS changes: either from mutation, perturbation to solution conditions, or by splitting FUS, as well as discussion of the meaning of such changes relative to co-partitioning of other key factors such as client proteins.

2. Reproducible trends in hydrodynamic radius (RH)? The authors demonstrate that the hydrodynamic radii of both FUS and PEG increase at higher $[1,6-HD]$ (Fig 5f, Fig S11). What about for FUS at varying low-end concentrations of KCl. Does hydrodynamic radius increase with increased $[KCL]$? They note an increased density of FUS in condensed phase at lower KCL which is consistent with a reduced hydrodynamic radius. They also suggest charge shielding by ions may lead to a larger hydration shell for FUS, which would be correlated with a higher $csat$, or weaker LLPS propensity. If so, does this fit a general trend that less collapsed chains have weaker homotypic interactions, increased $csat$ s, and altered dominance? Or is dominance not necessarily linked to RH?

3. PEG partitioning versus crowding function. I was a bit confused by how to interpret the data in Figure 5 with respect to the ‘collective interactions’ driving FUS condensation. PEG is used as a crowding agent – reducing the available solvent - but the authors also describe its partitioning, varying from 0.69 (w/v PEG 10000 / μM FUS) to 0.38 upon addition of 1,6-HD. The radius of hydration of PEG and FUS also increased in presence of 1,6-HD. What is unclear to me is whether altered RH or FUS and PEG in presence of 1,6-HD affects their collective interactions, or has a more direct effect on crowding? How can the authors differentiate between these effects?

Minor: As a control, can they utilize a second non-ionic small molecule that co-partitions with FUS to determine its effect on dominance. For example: BODIPY or Nile red. These would only be expected to strongly alter FUS dominance at re-entrant high salt regime. And have little effect on FUS condensation at for example 50 mM KCl

Version 1:

Reviewer comments:

Reviewer #1

(Remarks to the Author)

The authors have done an exceptional job addressing all my questions. I do not have any other concerns. This is a great work and is now ready for publication.

Reviewer #3

(Remarks to the Author)

In this revised and improved manuscript the authors have address my major concerns. In particularly, by expanding the scope of the study to TDP-43 and PGL-3 they have provided evidence for reproducibility of the trends independent of polypeptide sequence, and shown that distinct sequences have difference dominant fractions. The addition of new illustrations further aided clarity.

I have one minor comment that needs to be addressed. I may have missed it, but I did not see the primary data for TDP-43 and PGL-3 measurements in the Supplementary Data File (corresponding to what was shown for FUS in Figs S3 and S4). The authors should absolutely include these as additional supplemental figures as they are essential for the reader.

I have also evaluated the authors' responses to Reviewer 2's comments, which can be found in the attached document.

This is an excellent overall study that I believe will be well-cited and the concepts and techniques will be an aid to the condensate community, providing a framework toward future characterization of heterotypic interactions with small molecules, RNAs and other peptides

Reviewer #4

(Remarks to the Author)

I was tasked by the editor to provide a technical assessment of the microfluidic approach used in this manuscript to ensure that the findings are valid and sound. Since the manuscript has already undergone revision, I am only providing new comments separate from what has already been mentioned in previous reviews.

Here, the authors utilize a microfluidic flow cell with confocal detection to quantify dilute phase concentrations, and thereby determine if and when biomolecular condensates have formed by phase separation. The overall microfluidic method and approach are well-described in the supplementary information document, and the general idea of using sudden spikes in fluorescence intensity--with the reasonable assumption that such spikes likely come from condensates--makes a lot of sense.

One area of potential clarification is that while it is expected that the baseline intensity would decrease when condensates have formed, this is not clearly seen in the data aside from the 20 mM KCl case in Fig. S2. I wonder if the authors can explain the anticipated amount of intensity decrease after condensate formation, and whether the amount measured is consistent with what they expect.

A minor suggestion would be to also include experimental confocal images of the condensates observed in the microfluidic flow cell, if those are available.

Otherwise, the microfluidic technique appears to be effective at detecting and quantifying condensate formation, and the manuscript overall contributes meaningfully to this field.

Version 2:

Reviewer comments:

Reviewer #4

(Remarks to the Author)

The authors have adequately addressed the minor comments and suggestions I made previously. I do not have any further concerns or suggestions.

Response to Reviewer's Comments

We very much appreciate the time and efforts made by the editor and the referees in reviewing this manuscript and gratefully acknowledge their positive and constructive feedback. We have addressed all issues raised by the reviewers and revised the manuscript accordingly in the light of their useful suggestions and comments. Crucially, this includes the generalisation of previous observations across a range of proteins as well as application of our approach to complex systems including intracellular data.

Reviewer #1 (Remarks to the Author):

In this manuscript, the authors have formulated a method for quantifying the molecular interactions involving all the components of a biomolecular condensate by measuring dilute or dense phase concentrations of only one component. By this, one can derive a quantitative and a collective information regarding all the components in the system such as stoichiometry, free energy of contribution, etc., by measuring phase boundaries and tie-lines of only one component (component A) and its dilute phase concentrations (C_{dil}^A) as a function of the second component. This method of analysis could be of very useful to estimate the boundary conditions of phase separation quantitatively to derive important characteristics of the condensate. Therefore, this manuscript can be highly impactful for the biomolecular condensate community. The authors may consider addressing the following to resolve or clarify a few issues that will help better the manuscript.

We thank the reviewer for their positive feedback, including the recognition of the potential impact our work could have on our community. We have aimed at extensively addressing all comments and believe this has aided further improving the manuscript.

It would be helpful for the reader if the derivation of relationship (Eq 1) is elaborated in methods or supplementary section:

$$D^A = \left[\frac{\Delta f}{\Delta f} \right]^A = K/(K-P) \text{ (Eq 1)}$$

According derivation has now been added in the methods section under 'Derivation of the dominance framework'.

The explanation for FUS undergoing LLPS in low ionic strengths is sound (page 4; paragraph 2) and well supported by the formulation and the models. However, this has to protein sequence-specific and cannot be generalized.

This is an excellent remark by the reviewer as the sequence-specificities of the individual protein will certainly affect phase behaviour. Having said that, many phase separation prone proteins share the propensity to undergo phase separation at low ionic strength. To further investigate this, we have repeated similar measurements with the pathologically and developmentally relevant intrinsically disordered proteins TDP43 and PGL3, respectively (see Response Fig. 1). Both proteins, similarly to FUS, display ion exclusion from the dense phase upon phase separation at low ionic strength, highlighting this as more generic physico-chemical mechanism underlying collective interactions. The sequence specificities referred to by the reviewer still play an important role as the c_{sat} , shape of the phase boundary, degree of ion partitioning, and the relative free energy gain of the protein vary for the individual sequences, but the ion exclusion mechanism appears to be conserved. Accordingly, this was incorporated in Figure 3 and the text, see p. 4, Line 148 ff.

Response Figure 1: Changes in dilute phase concentration with varying KCl concentrations for FUS, TDP43 and PGL3. All three proteins display a negative reduced tie line gradient ($K < 0$). While their differences in sequence specificity affect the c_{sat} , dominance and degree of partitioning, the underlying mechanism driving interactions is conserved.

The data in the manuscript soundly demonstrates the methodology works for a binary component system (FUS and salt). However, it would greatly help if the authors could demonstrate/comment on how these will be helpful for multi-component systems.

We thank the reviewer for pointing to the important context of complex condensation. Crucially, our approach is specifically tailored for applicability independent of the system complexity as dilute phase concentration measurements of an individual component suffice to obtain mechanistic information. To highlight this further, we have applied our approach to intracellular dilute phase quantification data from literature¹, showing applicability even *in cellulo*. In this data set, we assess the differences in the collective protein interactions between opto-protein variants and even G3BP1. For detailed discussion please refer to Response Figure 2 below and manuscript p. 8, where we have incorporated a new section ‘Translation to complex environments’ based on this data. Together, this example highlights the applicability and simply translation of our approach to phase separation under complex environments.

Response Figure 2. Intracellular dominance determination. (a) Homotypic line scans are performed by quantifying dilute phase concentrations of component A while varying its total concentration. (b) Homotypic response function of G3BP1 *in cellulo* from ¹. (c) Comparison of homotypic response functions of opto-protein fusion constructs intracellularly from ¹. (d) Dominance fraction comparison of different proteins in (c).

Can the authors comment on why the slopes of $C_{\text{dil}}^{\text{FUS}}$ line scans at low salt concentrations are linear while they are non-linear for re-entrant phase separation at high ionic strengths? Can a second ‘R’ value be calculated for re-entrant conditions and what would it inform us?

The low- and high-salt regimes have different phase separation mechanisms leading to different responses to the addition of ions. In the low-salt regime, the electrostatic attraction at zero salt is an upper bound on the protein-protein attraction that one cannot surpass by decreasing the amount of salt. On the other hand, in the high-salt regime, hydrophobic attraction is strengthened if additional salt is added, and this does not seem to hit a hard bound. This is also supported by the concentration range required to see sufficient differences in the dilute phase response, which is significantly greater in the high ionic strength regime.

In the determination of R, wouldn't it be better to average slopes of three line-scans than two?

This is an excellent observation by the reviewer which would theoretically lead to a more accurate characterisation. In this work, however, we specifically focus on employing the simplest possible measurement approach for which two scans are sufficient to obtain desired information. A comment was added to the main text on page 2, line 86, referring to the potential added value of measuring more of these scans.

Since reduced tie-lines within the demixed phase describe compositions along which the dilute phase concentrations of a component does not vary, it is unclear from Fig 3c, as to what is indicated as C_{tot} and C_{dil} along the tie-line.

We thank the reviewer for the useful point and have removed these annotations and amended the figure to include an arrow to indicate the trajectory along the reduced tie line.

Minor points

In figures 2-5, it would be good to indicate what the color codes for circular symbols mean in the legends. As is, it remains an interpretable aspect.

We thank the reviewer for this remark. As prompted, we have included a legend in Figure 2, and removed the colours from Figures 3-5 as the primary purpose was to visually aid the explanation of the experimental approach.

Figures 4d,e & f are not cited in the text.

References to figures 4d, 4e, and 4f have now been placed in the text (Lines 186, 19 and 205).

Author contributions for AAH, and SA are not mentioned.

This has now been corrected and the contributions for AAH and SA have been included.

Reviewer # 2 (Remarks to the Author):

In this work, Ausserwöger et al. present a new strategy to quantify collective interactions in biomolecular phase separation. By performing line scans on samples with a constant concentration of fluorescently labeled component A (FUS) and varying concentrations of component B (e.g. KCl), phase separation of component A into a dense phase can be quantified through detection of a spike in fluorescence. From this data, the phase boundary between components A/B is determined, and key parameters such as the tie line gradient K and phase boundary gradient P are extracted. Finally, the dominance D of component A can be determined using K and P , quantifying the relative contribution of A to the free energy decrease of phase separation. The authors suggest that intracellular chloride ions may modulate FUS condensates in vivo. However, they only present in vitro data, thus applicability in vivo remains unknown. Similarly, the insight into the behavior of 1,6-hexandiol as a solvating agent is interesting, but their conclusions are supported with observations of FUS interactions/condensates, so it is not clear if that observation is generalizable to other protein condensates. Overall, while this is an interesting and potentially useful method, it is unclear the findings from the FUS system in vitro are generalizable to other condensates and in vivo. Further, some of the conclusions seem overstated, given the lack of experimental evidence, I cannot recommend publication in Nature Communications.

We thank the reviewer for their assessment of our work and raising their potential concerns with regards to the translatability of our findings to other protein systems. To systematically and conclusively address these comments, we have repeated key measurements with additional protein systems including PGL3 and TDP43. These measurements confirm the potential generic mechanism of ion exclusion at low ionic strengths as well as the 1,6-hexandiol polymer expansion on all other probed systems. Detailed

discussion of the added data and observations is provided below around Response Figures 3 and 4. Furthermore, we would like to note, that we had previously only mentioned the potential impact of our observations on intracellular FUS phase separation in a brief paragraph in the discussion to provide some further context to the potential meaning of our discoveries. To alleviate the reviewers concerns around this, we have now largely removed this part to focus on fundamental physico-chemical mechanistic insight gained and the technological advancements provided. Additionally, in Figure 6 (Response Figure 5 here) and associated text section we have now also included the translation of our approach to intracellular data. In doing so, we demonstrate direct applicability of our strategy to quantify collective interactions also *in cellulo*.

1. The data shown on FUS and ion concentrations is consistent with the proposed mechanism (i.e. intracellular chloride ions may modulate FUS condensates *in vivo*), but the conclusions in general are overstated. The authors observe correlations with the ion additions, but have no evidence for the molecular mechanism behind their macroscopic observation. Furthermore, it is unclear how generalizable the observations for effects of ions and hexanediol on FUS phase behavior are to other proteins, with other surface charge densities and conformations. In lieu of this, the authors should tamp down their claims, or provide evidence for the proposed mechanism as well as evidence for the observed behavior for other proteins.

We are pleased to share that, prompted by this comment from the reviewer, we have expanded our investigations to other homotypic phase separation-prone proteins (TDP43, PGL3). Crucially, we observe that both the ion exclusion and hexanediol expansion mechanisms are conserved between this set of proteins (see Response Figure 3 and 4 below), indicating generalisability of the observed behaviour. Specifically, both TDP43 and PGL3 also display exclusion of ions from the dense phase as shown by a negative reduced tie line gradient ($K < 0$). Similarly, PGL3, TDP43 and SOX2 all display polypeptide chain expansion upon addition of 1,6-HD. Accordingly, this data has been added to the manuscript to expand and support the proposed modulation mechanisms (Ion exclusion: Figure 3, p.4; Hexanediol: Figure 5, p.8).

Regarding the suggested lack of evidence for the molecular mechanism of the ion release behaviour, we kindly point the reviewer to our data in Figure 3 highlighting a negative reduced tie line gradient, the relative free energy gain of FUS and the fluorescence lifetime measurements which all coherently corroborate the proposed mechanism. Furthermore, we have also pointed out in detail other studies that similarly indicate the proposed partitioning behaviour. This includes simulation results for FUS specifically² as well characterisation of polyelectrolyte systems including direct quantification of the dense phase concentrations^{3,4}. Quantification of ion partitioning directly in protein condensates is, so far, largely inaccessible, hence, requiring new methodologies such as our approach to measure.

As mentioned in our initial response to the reviewer, we have also adapted the conclusion by removing the discussion around the effect of ions in the intracellular environment on p. 9 to focus on the physical chemistry.

Response Figure 3: Changes in dilute phase concentration with varying KCl concentrations for FUS, TDP43 and PGL3. All three proteins display a negative reduced tie line gradient ($K < 0$). While their differences in sequence specificity affect the c_{sat} , dominance and degree of partitioning, the underlying mechanism driving interactions is conserved.

Response Figure 4: Changes in the protein hydrodynamic radius with increasing hexanediol concentrations for FUS, TDP43, SOX2 and PGL3. Hexanediol dissolution effects on all these proteins have been established previously⁵.

2. Lines 317-318 are overstatements that require more disclaimers or experimental evidence for this behavior in vivo. The causative agent for ion release has not yet been determined, and this in vivo relevance depends entirely on the ions driving phase separation behavior of FUS. There may be any number of in vivo mechanisms that turn this in vitro behavior on its head. For example, how would the cell pumping of ions against the concentration gradient affect this behavior?

We thank the reviewer for pointing out their concerns and would like to highlight that our efforts to discuss this effect were merely aimed at pointing out fluctuations of the intracellular chloride ion concentration as a potential lever of interest for the behaviour of phase separation-prone proteins. This comes in light of the pronounced effect of ions on protein phase behaviour, an observation repeatedly made in the literature, and now further corroborated by our data. Accordingly, this discussion was refined to represent this more clearly and was overall amended to focus on the fundamental physico-chemical mechanisms, see p.9.

3. The use of the phrase “collective interactions” should be reconsidered. While this method does report on the interactions between different components in a two-component phase separated system, it actually describes the macroscopic phase separation, so a better term would be “bulk interactions.” The term “collective interactions” is confusing as authors do not show any emergent collective behavior, but only refer to pair-wise interactions.

We thank the reviewer for allowing us to further elaborate on the usage of the terminology ‘collective interactions’. Despite the reviewer’s comment, our approach very distinctly does not refer to pair-wise interactions. For example, we quantify component partitioning which can only occur upon the formation of a distinct phase as a consequence of a set of interactions between a large number of molecules. The formation of an ion gradient, for example, illustrates an emergent behaviour that can only arise from a collective interaction. Pair-wise binding between individual molecules cannot yield such behaviour, thus necessitating the reference to the collective properties characterising these interactions. While we appreciate the reviewers suggestion to utilise ‘bulk interaction’, we note the importance of differentiating between the bulk dense and dilute phase, hence, necessitating further specification.

4. Is there a reason why only low salt concentration with KCl and high salt concentration with LiCl and CsCl was studied, but no low concentration with LiCl? The varying effect on FUS was explained via the high ionic strength of LiCl/CsCl vs. KCl, so it would strengthen the claim by confirming this varied behavior at low salt concentrations of LiCl and CsCl.

These experiments were not previously carried out due to technical limitations regarding the stability of the protein in buffers at low ionic strength making it challenging to vary the primary counter ion without expansive further effort. Having said that, we do believe that the addition of further protein systems has helped expand and substantiate this section significantly.

5. It is mentioned that the dilute and dense phase concentrations are quantified using the intensity readouts from the linescans. How does this procedure work? The total concentration of protein is known, but how is the ratio of dilute to dense phase calculated based on the fluorescence readouts? Is photobleaching accounted for?

This is a really important point raised by the reviewer as our work specifically focuses on measuring dilute phase concentrations only. Although also stated before, we have now aimed at further clarifying this in the text. In our method, the concentration of the dilute phase is extracted by recording the baseline fluorescence intensity of phase separated samples in a small confocal volume (~1x1um) under flow. Here, continuous replenishment of the sample excited in the confocal volume paired with minimal excitation times and laser powers mitigate any possibility of photobleaching affecting the signal readout. It is further worth noting that any other quantification method of the dilute phase concentration would be compatible with the framework rendering the approach highly flexible experimentally.

6. P. 2, line 77: The authors state that this approach could be combined with standard epifluorescence or absorbance-based measurements. How would standard epifluorescence work, given that it integrates the fluorescence signal in z? Would it not only read out the total fluorescence of the dilute and dense phase, leading to the same result in every case?

This is an excellent observation by the reviewer. In the sentence referred to, we had previously mentioned that these approaches would need to be combined with a method to separate the dense and dilute phases, such as centrifugation. We have also revised these statements to further enhance clarity.

7. The confocal microscope description shows two wavelengths used, both 485nm and 640nm together using Pulsed-Interleaved Excitation. The work describes using GFP tagged polymers, which explains the 485nm laser. I was not able to find what fluorophore is the 640nm laser exciting, and what is its purpose? Also, while it is implied from the supplementary figure, it should be stated explicitly whether this is a scanning or spinning disk confocal microscope in the Methods section.

We thank the reviewer for pointing this, while the 640nm laser line is present in the set-up, it was not used here. The scheme has been updated and it is also now explicitly stated that this is a scanning confocal microscope.

8. Is there mixing of the prepared solutions upon loading into the channel? The SI states “following equilibration, dilute phase concentration were measured...” but this should be expanded to explicitly consider mixing. Is there a control for avoiding heterogeneity in the system due to bad mixing, as opposed to phase separation?

Samples were mixed off-chip to avoid sample heterogeneity or other issues caused by suboptimal mixing. Effective mixing is further confirmed here by a stable baseline fluorescence signal during acquisition where the continuous flow leads to sampling of a significant fraction of the total sample volume. An additional note on off-chip mixing was now added to the Methods section.

9. The hydrodynamic radius of FUS-EGFP measured for the hexanediol experiments is helpful. To support authors' claims about the behavior of ions, the effect of ions on the hydrodynamic radius of FUS-EGFP should also be quantified. Overall, the interpretation about the effect of ions is a bit of a stretch, given the experimental evidence presented. While the interpretation is feasible, the authors should provide an orthogonal measurement allowing to assess the changes in intra-FUS distance, hydrodynamic radius, etc in the presence of salts.

Quantifying the protein expansion as a function of the environmental ion concentration is an excellent suggestion by the reviewer, which we have accordingly implemented. Interestingly, we only see minimal changes in the FUS hydrodynamic radius with decreasing ionic strength. However, when considering the physico-chemical properties of the FUS intrinsically disordered region, we find that it contains a very low fraction of charged residues. Hence, the protein expansion associated with the disordered domain, is not affected much by the ion concentration in this case. The intermolecular

electrostatic attraction and repulsion, however, can still be affected by the structured domains of FUS. For comparison we have performed the same experiment with G3BP1, which is known for high charge content in its IDRs. Specifically, G3BP1 displays blocks of positive and negative charge in the IDRs to achieve a self-inhibiting conformation. Accordingly, when decreasing the ionic strength G3BP1 shows substantial compaction as electrostatic intramolecular interactions become more favourable in the protein IDRs. Conversely, G3BP1 actually does not show size expansion under the influence of 1,6-HD, whereas FUS and other proteins do. This can be traced back to the electrostatic self-inhibitory confirmation of the protein, which is primarily affected by the ionic strength. Please refer to Response Figure 5.

Response Figure 5. Polypeptide chain expansion under low ionic strength conditions. (a) Hydrodynamic radius change comparison between FUS and G3BP1 at low ionic strength conditions. (b) Sequence charge fraction comparison between FUS and G3BP1 as well as their disordered regions. (c) Illustration of the G3BP1 IDR sequence charge as reproduced from ⁶ (top) and ⁷ (bot). (d) Hydrodynamic radius change of G3BP1 under change of the 1,6-HD concentration.

10. Lines 173-175, the observation that ion release triggers phase separation is overstated. A correlation of these two states was observed, but causation was not determined. The sentence should be reworded, or additional direct measurements should be reported to support this claim.

We thank the reviewer for pointing this out and agree that the statement could benefit from rewording. Specifically, the decrease in ionic strength acts as a trigger for allowing phase separation to become spontaneously favourable. Here, ion exclusion from the dense phase and a release of ions from the polypeptide chains are significant contributors to overall driving force of phase separation. We have amended the phrasing accordingly (Lines 146 and 147).

11. Line 190, strong non-ionic interactions arising from an increase in ionic strength is mentioned. Could authors list, which non-ionic interactions they are considering here?

The driving forces for high salt reentrant phase separation have previously been shown in detail to include non-ionic interactions such as pi-stacking or non-polar contacts as well as hydrophobic interactions driven by interfacial water release⁵. The statement was expanded to provide more detail discussion (Line 173ff).

12. Line 326, the authors mention “a multitude of commonly available measurement assays”, but when introducing their method, they only list epi-fluorescence and absorbance. Are there any more? Please list. Alternatively, remove “multitude” as it evokes a high number of possible approaches.

We thank the reviewer for raising this comment and have changed ‘a multitude of commonly available measurement assays’ to ‘using commonly available measurement assays’ (Line 345)

13. The presented framework relies on a classic equilibrium thermodynamics. Have the authors considered entropy and enthalpy connected with such phase separation and its possible effect in vivo?

Entropy and enthalpy effects are both contained in the general free energy $f(\phi)$ used in the dominance framework. Qualitatively, the entropic effect dominates at very low protein concentrations due to the logarithmic scaling of the chemical potential originating from translational entropy, and enthalpy is

more important at higher protein concentrations due to its polynomial scaling. A detailed discussion of how entropy and enthalpy fit in the dominance framework requires introducing mathematical models of phase separation (Flory-Huggins for instance) and analysing their phase space using the dominance approach, which is quite beyond the scope of the current manuscript. In cells, active processes could potentially change this equilibrium picture, but it is possible to treat them as cases of dynamic equilibrium, so the proposed framework remains applicable.

14. The applicability of these observations to *in vivo* systems is unclear, due to many obvious differences between the systems. Beside activity, etc, liquid condensates *in vivo* contain RNA as ubiquitous ingredient, with RNA being charged, flexible, etc, hence very likely influencing the effect of salts on proteins in condensates *in vivo*. Thus, it is difficult to draw conclusions about phase behavior of liquid condensates *in vivo*.

We appreciate the opportunity to discuss this work in the context of complex systems, such as within cells. As the reviewer rightly pointed out, the diversity of biomolecules and interactions introduces additional factors to consider. However, the fundamental physico-chemical principles driving protein interactions, such as the release of part of the ionic hydration shell to enable intermolecular interactions, are unlikely to be rendered completely irrelevant even in more complex systems.

That said, we agree with the reviewer that it is crucial to also study the phase behaviour of condensates *in vivo*. To address this, we reanalysed *in cellulo* datasets from the literature, demonstrating that our approach is applicable to complex environments. In doing so we can assess energetic differences in the collective interactions between opto-protein variants and G3BP1. For detailed discussion, please refer to Response Figure 2 below and manuscript p. 8, where we have added a new section titled 'Translation to complex environments' based on this data.

In summary, understanding the fundamental physico-chemical processes driving protein interactions is key to unravelling the complexities of intracellular environments. Our approach, as supported by the additional data, enables this characterisation directly in cells.

Response Figure 6. Intracellular dominance determination. (a) Homotypic line scans are performed by quantifying dilute phase concentrations of component A while varying its total concentration. (b) Homotypic response function of G3BP1 *in cellulo* from ¹. (c) Comparison of homotypic response functions of opto-protein fusion constructs intracellularly from ¹. (d) Dominance fraction comparison of different proteins in (c).

Minor Points:

1. Line 25 typo: “would enable to address”

This has been corrected.

2. Figure 4e and 4f are discussed but not explicitly referenced in the text anywhere.

Figures 4e and 4f are now referred to in the text at the according positions (Lines 199 and 205).

3. Figure S3, subplots are not labelled, and referred to as “a”, “b”, and “left panel”.

This has been corrected.

4. Supplementary Figure S2 is labeled as S1, and all subsequent SI figure titles are off by 1.

We thank the reviewer for pointing this out, which has been amended accordingly.

5. Figure 4a: “reentrant phase transition”

Corrected.

6. Figure 4c: “no-ionic interactions” should be “non-ionic interactions.”

Yes, correct – amended. Thank you.

Reviewer # 3 (Remarks to the Author):

In this manuscript the author develop an experimental and theoretical framework in an attempt to better quantify interactions among polypeptides and small molecules that shape the landscape of biomolecular condensation. Their strategy is to understand liquid-liquid phase separation (LLPS) by characterizing the behavior of tie lines and component dominance, or fractional free energy decrease from one component relative to the overall free energy decrease from phase separation. To do so they utilize a microfluidic system to vary the concentration of a model disordered scaffold protein, FUS, along with salts and hydrophobic molecules, and measure the concentration of the FUS in the dilute phase using fluorescence confocal microscopy. By varying salt, KCl, concentration at two different FUS-EGFP concentrations they are able to determine saturation concentrations and tie lines to calculate the parameters P, K which can ultimately be used to calculate fractional dominance DA for the scaffold or one of other components. Using such a setup they first determine that the dominance of FUS-EGFP is 0.61, meaning ~ 61% of the total free energy decrease from LLPS can be attributed to FUS. The other portion is due to complex interaction with ions and water. Next, they characterize the effect on FUS density and fluorescence lifetime in the condensed phase as a function of low KCl concentrations. As previously known in the field, increased [KCl] blocks FUS condensation, with a re-entrant LLPS phenomenon at very high concentrations. By mapping the dilute phase concentration of FUS relative to [KCl] they determine a reduced negative tie line gradient ($K < 0$) indicative that the presence of KCl blocks FUS condensation. This trend indicates that KCL tends to be excluded from FUS condensates. Additionally, measurements of FUS-EGFP lifetime in dense phase that decreases at lower [KCl], suggesting that dense phase becomes more packed at lower salt. Intriguingly at very high salt concentrations (5M LiCl), FUS displays re-entrant phase separation driven by non-ionic interactions. The authors also are able to extrapolate that LiCl enriches in FUS condensates under these conditions at a ratio of ~ 140 mM ion per micromolar of FUS. Finally, they explore the extent to which 1,6 hexanediol (1,6,-HD) impacts FUS LLPS in the presence of PEG 10000, and FUS dominance. They propose that addition of 1,6-HD raises csat, lowers PEG partitioning and increases fractional dominance.

Overall, this is a clear and quantitative study of the parameters that contribute to collection interactions that drive disordered protein phase separation, authored by a number of leaders in the field. Strengths of the work include careful measurements of dilute phase FUS concentrations as a function of ions and other small molecules, calculation of tie lines and csats to determine the dominance of FUS, and illustrative diagrams and well written text to explain the trends in phase diagrams. Enthusiasm for the work is somewhat dimmed by the limited experimental nature of the study; the authors limit their characterization to well-defined phenomena. I have three major suggestions for improving the manuscript below

We thank the reviewer for their supportive summary of our work and appreciate their valuable suggestions. We have taken the points raised by the reviewer on board and have collected additional data to present in the manuscript including protein independent generalisation of the salt-exclusion and hexanediol protein expansion mechanisms, Rh change/phase separation correlation, as well as application of our approach to in cellulose data. We further would like to note that while phase separation

at low ionic strengths and dissolution via hexanediol are well defined phenomena, the mechanistic insights gained through our quantification approach have so far not been addressed in the field. This includes partitioning of ions, connection between the protein expansion and phase separation.

1. Significance or meaning of changes to fractional dominance of FUS. The authors calculate dominance of FUS as between ~ 0.6-0.7, dependent on the presence of KCl, PEG, 1,6-HD. However, it would be very helpful to additional examples in which fractional dominance shifts more significantly. What about use of a FUS mutant that has a higher csat (such as from mutation of Arg or Tyr) or lower csat (from addition of Tyr). Would this be expect to alter the dominance fraction? If it alters the hydrodynamic radius this in fact may be the case. Additionally mutation of Arg would affect interactions with ions. Alternatively, the authors should perturb the setup in some other fashion: altering pH would affect ionizable groups and interactions with KCl; they might test other types of salts. Framed another way, is there a theoretical reason to expect for a disordered protein with a csat of ~ 2 μ M that its dominance will never drop much below 0.5-0.6. Can they provide a simulation to support this. Alternatively, for context, what is dominance is a polyelectrolyte coacervate system: of polyanion and polycation; in this case is the fractional dominance of each ~ 0.35 and the ions also 0.3-0.4? If so, can the authors test a split version of FUS, into N-terminal FUS LC and and C-terminal fragment that some of the authors previously demonstrated forms complex coacervates (Wang and Hyman, Cell 2018). In short, the manuscript would be greatly strengthened by provided additional cases in which dominance of FUS changes: either from mutation, perturbation to solution conditions, or by splitting FUS, as well as discussion of the meaning of such changes relative to co-partitioning of other key factors such as client proteins.

We thank the reviewer for their comment and the detailed suggestions and accordingly we have added dominance quantification data across a range of protein systems and solution conditions.

First, we would like to point towards our data and discussion around phase separation of FUS at high ionic strength. This scenario provides quite detailed insights into the extreme cases for the protein dominance. Specifically, at high protein concentrations (> 2 μ M) the system is purely limited by the available ion concentration leading to a phase boundary gradient \rightarrow Inf, i.e. parallel to the FUS-axis. Reminding ourselves of the definition of the dominance:

$$D^A = \frac{\Delta f^A}{\Delta f} = \frac{K}{K - P}$$

this will give a dominance of 0, as the free energy contribution of the salt massively outweighs that of the protein. Conversely, at lower protein concentrations, the limiting scenario shifts effectively reversing the behaviour and leading to $D \sim 1$. Hence, extreme cases in the dominance can be found in scenarios where the relative component presence leads to limiting cases for driving the phase transition. To emphasise this further and make this more accessible we have amended Figure 4 accordingly, to highlight and explain the distinct extreme cases in panel (e).

That said, we further agree that it would be additive to probe the effect of changing the saturation concentration on potential variations in the dominance. The intracellular data around the opto-FUS system added as part of the revision provides an excellent example. The WT opto-FUS protein alone actually gives a dominance value close to $D \sim 1$, indicating that the phase behaviour is entirely driven by homotypic protein interactions, despite the complex cellular environment. The opto-FUS(5Y>S) variant, displays a drastic increase in the saturation concentration (~5x higher) indicating a weakening of the underlying interactions (Response Figure 7). The dominance, however, does not vary for this variant ($D \sim 1$) indicating that while the interactions have weakened, the phase transition is still purely driven by the protein interactions. Hence, while FUS/FUS contacts are less favourable upon removing critical tyrosine residues, the mode of assembly is still purely driven by these heterotypic interactions in this case. This indicates that a change in the saturation concentration does not have to coincide with a variation in the relative free energy contributions, particularly in extreme cases.

Further variation of the dominance can also be observed upon comparing phase separation of FUS, TDP43 and PGL3 under similar ionic strength conditions. Here, while the ion exclusion behaviour is observed for all three proteins, the energetic dependencies vary as dominances between 0.28 and 0.72 are observed (Response Figure 8). To establish a detailed connection between protein sequence and dominance, however, more rigorous investigations of large range of proteins will be required, which would necessitate a distinct investigation on its own.

Taken together, we now supply with our manuscript a range of examples of protein dominances at varying solution conditions and protein sequences as well as a dedicated discussion of the extreme cases. Hence, we believe our work now provides extensive data sets for readers to compare with and incentivise testing on their own systems, therefore, improving our work.

Response Figure 7. Intracellular dominance determination. (a) Homotypic line scans are performed by quantifying dilute phase concentrations of component A while varying its total concentration. (b) Homotypic response function of G3BP1 *in cellulo* from ¹. (c) Comparison of homotypic response functions of opto-protein fusion constructs intracellularly from ¹. (d) Dominance fraction comparison of different proteins in (c).

Response Figure 8: Changes in dilute phase concentration with varying KCl concentrations for FUS, TDP43 and PGL3. All three proteins display a negative reduced tie line gradient ($K < 0$). While their differences in sequence specificity affect the c_{sat} , dominance and degree of partitioning, the underlying mechanism driving interactions is conserved.

2. Reproducible trends in hydrodynamic radius (RH)? The authors demonstrate that the hydrodynamic radii of both FUS and PEG increase at higher [1,6-HD] (Fig 5f, Fig S11). What about for FUS at varying low-end concentrations of KCl. Does hydrodynamic radius increase with increased [KCL]? They note an increased density of FUS in condensed phase at lower KCL which is consistent with a reduced hydrodynamic radius. They also suggest charge shielding by ions may lead to a larger hydration shell for FUS, which would be correlated with a higher c_{sat} , or weaker LLPS propensity. If so, does this fit a general trend that less collapsed chains have weaker homotypic interactions, increased c_{sats} , and altered dominance? Or is dominance not necessarily linked to RH?

We thank the reviewer for this excellent suggestion and agree that the hydrodynamic radius / phase separation propensity correlation could similarly be relevant at low ionic strengths. To address this question, we have performed additional measurements at varying KCl concentrations (see Response Figure 9).

We find that the FUS hydrodynamic radius displays little variation when changing the KCl concentration. However, upon inspecting the physico-chemical properties of the FUS intrinsically disordered regions, it becomes apparent that these contain a very low fraction of charged residues, i.e. most charges sit on the structured domains. Hence, the protein expansion, which is largely governed by the disordered domains, is not affected much by the ion concentration in this case. The intermolecular electrostatic attraction and repulsion, however, can still be affected by the charges exerted through the structured domains of FUS, which similarly holds true for the density within condensates.

For comparison we have performed the same experiment with G3BP1, which is known for high charge content in its IDRs. Specifically, G3BP1 displays blocks of positive and negative charge in the IDRs to achieve a self-inhibiting conformation. Accordingly, when decreasing the ionic strength, G3BP1 shows substantial compaction as electrostatic intramolecular interactions become more favourable in the protein IDRs. Conversely, G3BP1 does not show size expansion under the influence of 1,6-HD or undergo homotypic phase separation, whereas FUS and other proteins do. This can be traced back to the electrostatic self-inhibitory conformation of the protein, which is primarily affected by the ionic strength or the presence of RNA^{6,7}.

As such, our data indicates that there appears to be a sweet spot in the IDR chain collapse to facilitate phase separation. At high compaction, polymer chains primarily engage in short ranged intramolecular interactions, thereby blocking intermolecular contacts. Very large expansions on the other hand indicate highly favourable polymer-solvent interactions, hence, rendering intermolecular interactions unfavourable.

Taken together, we believe that there likely is a correlation between the IDR expansion behaviour and the physico-chemical driving forces for phase separation given the underlying polymer physics. However, the distinct sequence specificities and diversity of proteins, as shown from the comparison between FUS and G3BP1, make a quantitative generalisation much more challenging. Because of that, we believe, while highly interesting, that establishing this correlation in detail necessitates measurements of large numbers of proteins including distinct sequence modulations which would warrant a distinct study to follow up on.

Response Figure 9. Polypeptide chain expansion under low ionic strength conditions. (a) Hydrodynamic radius change comparison between FUS and G3BP1 at low ionic strength conditions. (b) Sequence charge fraction comparison between FUS and G3BP1 as well as their disordered regions. (c) Illustration of the G3BP1 IDR sequence charge as reproduced from ⁶ (top) and ⁷ (bot). (d) Hydrodynamic radius change of G3BP1 under change of the 1,6-HD concentration.

3. PEG partitioning versus crowding function. I was a bit confused by how to interpret the data in Figure 5 with respect to the ‘collective interactions’ driving FUS condensation. PEG is used as a crowding agent – reducing the available solvent - but the authors also describe its partitioning, varying from 0.69 (w/v PEG 10000 / μ M FUS) to 0.38 upon addition of 1,6-HD. The radius of hydration of PEG and FUS also increased in presence of 1,6-HD. What is unclear to me is whether altered RH or FUS and PEG in presence of 1,6-HD affects their collective interactions, or has a more direct effect on crowding? How can the authors differentiate between these effects?

We thank the reviewer for pointing out the need for further clarification of the behaviour of PEG in this context.

We first evaluated the collective properties of the FUS-PEG system in absence of 1,6-HD. In doing so, we find a positive reduced tie line gradient ($K > 0$), suggesting that PEG co-partitions into FUS condensates rather than acting as a traditional crowder. Hence, PEG can be viewed more so as a copolymer that scaffolds interactions in this context rather than excluding solvent, which has also been observed before, even for other proteins^{8,9}.

Based on this observation, we then proceeded to the addition of 1,6-HD to the FUS-PEG system and evaluated the relative change in the collective interactions. To present this more clearly, we have amended the text to first specifically discuss the FUS-PEG system behaviour in absence of 1,6-HD and then characterised the effect of the addition of the modulator. Please refer to Line 211 ff on p. 6.

Minor: As a control, can they utilize a second non-ionic small molecule that co-partitions with FUS to determine its effect on dominance. For example: BODIPY or Nile red. These would only be expected to strongly alter FUS dominance at re-entrant high salt regime. And have little effect on FUS condensation at for example 50 mM KCl

While this is another very useful suggestion by the reviewer, we have implemented the 1,6-HD as a specific pointer towards small molecule modulation behaviour. As such we feel a detailed evaluation of a range of small molecules extends beyond the scope of this work, which is focuses on methodological advancements as well as our physico-chemical understanding of collective interactions.

References

1. Riback, J. A. *et al.* Composition-dependent thermodynamics of intracellular phase separation. *Nature* **581**, 209–214 (2020).
2. Welsh, T. J. *et al.* Surface Electrostatics Govern the Emulsion Stability of Biomolecular Condensates. *Nano Lett* **22**, 612–621 (2022).
3. Li, L. *et al.* Phase Behavior and Salt Partitioning in Polyelectrolyte Complex Coacervates. *Macromolecules* **51**, 2988–2995 (2018).
4. Friedowitz, S. *et al.* Looping-in complexation and ion partitioning in nonstoichiometric polyelectrolyte mixtures. *Science Advances* **7**, eabg8654.
5. Krainer, G. *et al.* Reentrant liquid condensate phase of proteins is stabilized by hydrophobic and non-ionic interactions. *Nature Communications* **12**, 1085 (2021).
6. Guillén-Boixet, J. *et al.* RNA-Induced Conformational Switching and Clustering of G3BP Drive Stress Granule Assembly by Condensation. *Cell* **181**, 346–361.e17 (2020).
7. Yang, P. *et al.* G3BP1 Is a Tunable Switch that Triggers Phase Separation to Assemble Stress Granules. *Cell* **181**, 325–345.e28 (2020).
8. André, A. A. M., Yewdall, N. A. & Spruijt, E. Crowding-induced phase separation and gelling by co-condensation of PEG in NPM1-rRNA condensates. *Biophysical Journal* **122**, 397–407 (2023).
9. Qian, D. *et al.* Tie-lines reveal interactions driving heteromolecular condensate formation. *bioRxiv* 2022.02.22.481401 (2022) doi:10.1101/2022.02.22.481401.

Response to Reviewer's Comments

We thank the editor and referees for their assessment of our revised manuscript and are delighted with their overall highly supportive and positive feedback. Below we have further addressed the additional remarks raised as part of the second revision.

Reviewer #1 (Remarks to the Author):

Referee: The authors have done an exceptional job addressing all my questions. I do not have any other concerns. This is a great work and is now ready for publication.

Response: We thank the reviewer for their positive feedback and are pleased by their suggestion of readiness for publication.

Reviewer #2 (Remarks to the Author):

Response: The responses provided to the comments raised by Reviewer 2 were assessed by Reviewer 3. As per the report provided, Reviewer 3 has clarified that all points raised by Reviewer 2 were either conclusively addressed or where not possible our clarifications are justified, and no significant concerns remain. We thank Reviewer 3 for their additional assessment of these points and welcome their conclusions that all concerns have been clarified. Lastly, we note that as highlighted by Reviewer 3, Response Figure 5 was indeed strictly only added for the Rebuttal, hence, not included in the manuscript (Reviewer 2, Remark 8).

Reviewer #3 (Remarks to the Author):

Referee: In this revised and improved manuscript the authors have address my major concerns. In particularly, by expanding the scope of the study to TDP-43 and PGL-3 they have provided evidence for reproducibility of the trends independent of polypeptide sequence, and shown that distinct sequences have difference dominant fractions. The addition of new illustrations further aided clarity.

Response: We are delighted that our additional experimental data and efforts have conclusively clarified and addressed the reviewer's concerns.

Referee: I have one minor comment that needs to be addressed. I may have missed it, but I did not see the primary data for TDP-43 and PGL-3 measurements in the Supplementary Data File (corresponding to what was shown for FUS in Figs S3 and S4). The authors should absolutely include these as additional supplementtal figures as they are essential for the reader.

Response: This is an excellent remark by the reviewer and the data has now been added accordingly to the Supplementary Information (Fig. S6 and S7).

Referee: I have also evaluated the authors' responses to Reviewer 2's comments, which can be found in the attached document.

Response: As discussed in the response to Reviewer 2 we are thankful for the comments provided by the reviewer.

Referee: This is an excellent overall study that I believe will be well-cited and the concepts and techniques will be an aid to the condensate community, providing a framework toward future characterization of heterotypic interactions with small molecules, RNAs and other peptides

Response: We thank the reviewer for their strong support of our work and agree that this could provide a valuable tool for the condensate community for a range of important challenges.

Reviewer #4 (Remarks to the Author):

Referee: I was tasked by the editor to provide a technical assessment of the microfluidic approach used in this manuscript to ensure that the findings are valid and sound. Since the manuscript has already undergone revision, I am only providing new comments separate from what has already been mentioned in previous reviews.

Response: We welcome the reviewer's additional technical assessment of the microfluidic approach presented here given the importance of this technique in our work.

Referee: Here, the authors utilize a microfluidic flow cell with confocal detection to quantify dilute phase concentrations, and thereby determine if and when biomolecular condensates have formed by phase separation. The overall microfluidic method and approach are well-described in the supplementary information document, and the general idea of using sudden spikes in fluorescence intensity--with the reasonable assumption that such spikes likely come from condensates--makes a lot of sense.

One area of potential clarification is that while it is expected that the baseline intensity would decrease when condensates have formed, this is not clearly seen in the data aside from the 20 mM KCl case in Fig. S2. I wonder if the authors can explain the anticipated amount of intensity decrease after condensate formation, and whether the amount measured is consistent with what they expect.

Response: We appreciate the reviewer's detailed analysis and the positive feedback regarding the clarity and validity of our measurement approach. The point raised about the baseline intensity shift used to determine the dilute phase concentration is indeed important. This subtle baseline drift is difficult to see in Fig. S2 because the y-axis range is set to accommodate the high-intensity condensate peaks.

To address this, we also provide intensity histograms (e.g., right panel of SI Fig. S3), where the baseline shift is clearly visible. This histogram-based approach is more effective than zooming in on the raw traces, because protein condensates typically occupy a very small volume fraction of the sample. As a result, most intensity data points reflect the dilute phase, making it straightforward to identify the baseline shift under different conditions. For this reason, we use the peak in the intensity histogram to quantify the dilute phase concentration.

Finally, the observed decrease in the dilute phase concentration—by a factor of two, for instance, when reducing the KCl concentration—is consistent with our expectations as the system moves deeper into the two-phase region.

Referee: A minor suggestion would be to also include experimental confocal images of the condensates observed in the microfluidic flow cell, if those are available.

Response: This is a helpful suggestion by the reviewer, but confocal images are unfortunately not available with the existing instrumentation given its technical layout.

Referee: Otherwise, the microfluidic technique appears to be effective at detecting and quantifying condensate formation, and the manuscript overall contributes meaningfully to this field.

Response: We thank the reviewer for the overall positive feedback and assessment of our work.

Reviewer # 2 (Remarks to the Author):

In this work, Ausserwöger et al. present a new strategy to quantify collective interactions in biomolecular phase separation. By performing line scans on samples with a constant concentration of fluorescently labeled component A (FUS) and varying concentrations of component B (e.g. KCl), phase separation of component A into a dense phase can be quantified through detection of a spike in fluorescence. From this data, the phase boundary between components A/B is determined, and key parameters such as the tie line gradient K and phase boundary gradient P are extracted. Finally, the dominance D of component A can be determined using K and P , quantifying the relative contribution of A to the free energy decrease of phase separation. **The authors suggest that intracellular chloride ions may modulate FUS condensates in vivo. However, they only present in vitro data, thus applicability in vivo remains unknown.** Similarly, the insight into the behavior of 1,6-hexanediol as a solvating agent is interesting, but their conclusions are supported with observations of FUS interactions/condensates, so it is **not clear if that observation is generalizable to other protein condensates.** Overall, while this is an interesting and potentially useful method, (1) it is unclear the findings from the FUS system in vitro are generalizable to other condensates and in vivo. Further, (2) some of the conclusions seem overstated, given the lack of experimental evidence, I cannot recommend publication in Nature Communications.

We thank the reviewer for their assessment of our work and raising their potential concerns with regards to the translatability of our findings to other protein systems. To systematically and conclusively address these comments, we have repeated key measurements with additional protein systems including PGL3 and TDP43. These measurements confirm the potential generic mechanism of ion exclusion at low ionic strengths as well as the 1,6-hexanediol polymer expansion on all other probed systems. Detailed discussion of the added data and observations is provided below around Response Figures 3 and 4. Furthermore, we would like to note, that we had previously only mentioned the potential impact of our observations on intracellular FUS phase separation in a brief paragraph in the discussion to provide some further context to the potential meaning of our discoveries. To alleviate the reviewers concerns around this, we have now largely removed this part to focus on fundamental physico-chemical mechanistic insight gained and the technological advancements provided. Additionally, in Figure 6 (Response Figure 5 here) and associated text section we have now also included the translation of our approach to intracellular data. In doing so, we demonstrate direct applicability of our strategy to quantify collective interactions also *in cellulo*.

R3: Reviewer 2's main concerns can be summarized as follows: (1) Are the findings generalizable to other disordered proteins, and (2) are the findings from in vitro reconstitution generalizable to a cellular context. In short, the revised manuscript does directly address these two major concerns by providing new experimental data. The authors extend their findings on component dominance to two additional IDPs, TDP-43 and PGL-3 (new Figure 3). Additionally the use an optogenetic clustering method to determine dominance in cells (new Figure 6).

The reviewer also states a variety of minor comments or questions that are largely technical in nature. In my opinion these have been reasonable addressed by the authors in their revised manuscript.

1. The data shown on FUS and ion concentrations is consistent with the proposed mechanism (i.e. intracellular chloride ions may modulate FUS condensates in vivo), but the conclusions in general are overstated. The authors observe correlations with the ion additions, but have no evidence for the molecular mechanism behind their macroscopic observation. Furthermore, it is unclear how generalizable the observations for effects of ions and hexanediol on FUS phase behavior are to other proteins, with other surface charge densities and conformations. In lieu of this, the authors should tamp down their claims, or provide evidence for the proposed mechanism as well as evidence for the observed behavior for other proteins.

We are pleased to share that, prompted by this comment from the reviewer, we have expanded our investigations to other homotypic phase separation-prone proteins (TDP43, PGL3). Crucially, we observe that both the ion exclusion and hexanediol expansion mechanisms are conserved between this set of proteins (see Response Figure 3 and 4 below), indicating generalisability of the observed

behaviour. Specifically, both TDP43 and PGL3 also display exclusion of ions from the dense phase as shown by a negative reduced tie line gradient ($K < 0$). Similarly, PGL3, TDP43 and SOX2 all display polypeptide chain expansion upon addition of 1,6-HD. Accordingly, this data has been added to the manuscript to expand and support the proposed modulation mechanisms (Ion exclusion: Figure 3, p.4; Hexanediol: Figure 5, p.8).

Regarding the suggested lack of evidence for the molecular mechanism of the ion release behaviour, we kindly point the reviewer to our data in Figure 3 highlighting a negative reduced tie line gradient, the relative free energy gain of FUS and the fluorescence lifetime measurements which all coherently corroborate the proposed mechanism. Furthermore, we have also pointed out in detail other studies that similarly indicate the proposed partitioning behaviour. This includes simulation results for FUS specifically² as well characterisation of polyelectrolyte systems including direct quantification of the dense phase concentrations^{3,4}. Quantification of ion partitioning directly in protein condensates is, so far, largely inaccessible, hence, requiring new methodologies such as our approach to measure.

As mentioned in our initial response to the reviewer, we have also adapted the conclusion by removing the discussion around the effect of ions in the intracellular environment on p. 9 to focus on the physical chemistry.

Response Figure 3: Changes in dilute phase concentration with varying KCl concentrations for FUS, TDP43 and PGL3. All three proteins display a negative reduced tie line gradient ($K < 0$). While their differences in sequence specificity affect the c_{sat} , dominance and degree of partitioning, the underlying mechanism driving interactions is conserved.

Response Figure 4: Changes in the protein hydrodynamic radius with increasing hexanediol concentrations for FUS, TDP43, SOX2 and PGL3. Hexanediol dissolution effects on all these proteins have been established previously⁵.

R3: The authors addressed this concern of generalizability by characterizing TDP-43 and PGL-3 in new Figure 3.

2. Lines 317-318 are overstatements that require more disclaimers or experimental evidence for this behavior in vivo. The causative agent for ion release has not yet been determined, and this in vivo relevance depends entirely on the ions driving phase separation behavior of FUS. There may be any number of in vivo mechanisms that turn this in vitro behavior on its head. For example, how would the

cell pumping of ions against the concentration gradient affect this behavior?

We thank the reviewer for pointing out their concerns and would like to highlight that our efforts to discuss this effect were merely aimed at pointing out fluctuations of the intracellular chloride ion concentration as a potential lever of interest for the behaviour of phase separation-prone proteins. This comes in light of the pronounced effect of ions on protein phase behaviour, an observation repeatedly made in the literature, and now further corroborated by our data. Accordingly, this discussion was refined to represent this more clearly and was overall amended to focus on the fundamental physico-chemical mechanisms, see p.9.

R3: The authors revised the text to limit overstating their claim.

3. The use of the phrase “collective interactions” should be reconsidered. While this method does report on the interactions between different components in a two-component phase separated system, it actually describes the macroscopic phase separation, so a better term would be “bulk interactions.” The term “collective interactions” is confusing as authors do not show any emergent collective behavior, but only refer to pair-wise interactions.

We thank the reviewer for allowing us to further elaborate on the usage of the terminology ‘collective interactions’. Despite the reviewer’s comment, our approach very distinctly does not refer to pair-wise interactions. For example, we quantify component partitioning which can only occur upon the formation of a distinct phase as a consequence of a set of interactions between a large number of molecules. The formation of an ion gradient, for example, illustrates an emergent behaviour that can only arise from a collective interaction. Pair-wise binding between individual molecules cannot yield such behaviour, thus necessitating the reference to the collective properties characterising these interactions. While we appreciate the reviewers suggestion to utilise ‘bulk interaction’, we note the importance of differentiating between the bulk dense and dilute phase, hence, necessitating further specification.

R3: The authors maintain that their terminology is appropriate. I see no major concern here.

4. Is there a reason why only low salt concentration with KCl and high salt concentration with LiCl and CsCl was studied, but no low concentration with LiCl? The varying effect on FUS was explained via the high ionic strength of LiCl/CsCl vs. KCl, so it would strengthen the claim by confirming this varied behavior at low salt concentrations of LiCl and CsCl.

These experiments were not previously carried out due to technical limitations regarding the stability of the protein in buffers at low ionic strength making it challenging to vary the primary counter ion without expansive further effort. Having said that, we do believe that the addition of further protein systems has helped expand and substantiate this section significantly.

R3: Not addressed. However the authors claim this is not technically feasible. I see no major concern here.

5. It is mentioned that the dilute and dense phase concentrations are quantified using the intensity readouts from the linescans. How does this procedure work? The total concentration of protein is known, but how is the ratio of dilute to dense phase calculated based on the fluorescence readouts? Is photobleaching accounted for?

This is a really important point raised by the reviewer as our work specifically focuses on measuring dilute phase concentrations only. Although also stated before, we have now aimed at further clarifying this in the text. In our method, the concentration of the dilute phase is extracted by recording the baseline fluorescence intensity of phase separated samples in a small confocal volume (~1x1um) under flow. Here, continuous replenishment of the sample excited in the confocal volume paired with minimal excitation times and laser powers mitigate any possibility of photobleaching affecting the signal readout. It is further worth noting that any other quantification method of the dilute phase concentration would be compatible with the framework rendering the approach highly flexible experimentally.

R3: Addressed. This was already described in the original manuscript. The authors provide additional clarification in their response.

5. P. 2, line 77: The authors state that this approach could be combined with standard epifluorescence or absorbance-based measurements. How would standard epifluorescence work, given that it integrates the fluorescence signal in z? Would it not only read out the total fluorescence of the dilute and dense phase, leading to the same result in every case?

This is an excellent observation by the reviewer. In the sentence referred to, we had previously mentioned that these approaches would need to be combined with a method to separate the dense and dilute phases, such as centrifugation. We have also revised these statements to further enhance clarity.

R3: Addressed. The authors corrected the text accordingly.

6. The confocal microscope description shows two wavelengths used, both 485nm and 640nm together using Pulsed-Interleaved Excitation. The work describes using GFP tagged polymers, which explains the 485nm laser. I was not able to find what fluorophore is the 640nm laser exciting, and what is its purpose? Also, while it is implied from the supplementary figure, it should be stated explicitly whether this is a scanning or spinning disk confocal microscope in the Methods section.

We thank the reviewer for pointing this, while the 640nm laser line is present in the set-up, it was not used here. The scheme has been updated and it is also now explicitly stated that this is a scanning confocal microscope.

R3: Addressed. The authors corrected the text accordingly.

7. Is there mixing of the prepared solutions upon loading into the channel? The SI states “following equilibration, dilute phase concentration were measured...” but this should be expanded to explicitly consider mixing. Is there a control for avoiding heterogeneity in the system due to bad mixing, as opposed to phase separation?

Samples were mixed off-chip to avoid sample heterogeneity or other issues caused by suboptimal mixing. Effective mixing is further confirmed here by a stable baseline fluorescence signal during acquisition where the continuous flow leads to sampling of a significant fraction of the total sample volume. An additional note on off-chip mixing was now added to the Methods section.

R3: Addressed. The authors corrected the text accordingly.

8. The hydrodynamic radius of FUS-EGFP measured for the hexanediol experiments is helpful. To support authors' claims about the behavior of ions, the effect of ions on the hydrodynamic radius of FUS-EGFP should also be quantified. Overall, the interpretation about the effect of ions is a bit of a stretch, given the experimental evidence presented. While the interpretation is feasible, the authors should provide an orthogonal measurement allowing to assess the changes in intra-FUS distance, hydrodynamic radius, etc in the presence of salts.

Quantifying the protein expansion as a function of the environmental ion concentration is an excellent suggestion by the reviewer, which we have accordingly implemented. Interestingly, we only see minimal changes in the FUS hydrodynamic radius with decreasing ionic strength. However, when considering the physico-chemical properties of the FUS intrinsically disordered region, we find that it contains a very low fraction of charged residues. Hence, the protein expansion associated with the disordered domain, is not affected much by the ion concentration in this case. The intermolecular electrostatic attraction and repulsion, however, can still be affected by the structured domains of FUS. For comparison we have performed the same experiment with G3BP1, which is known for high charge content in its IDRs. Specifically, G3BP1 displays blocks of positive and negative charge in the IDRs to achieve a self-inhibiting conformation. Accordingly, when decreasing the ionic strength G3BP1 shows substantial compaction as electrostatic intramolecular interactions become more favourable in the

protein IDRs. Conversely, G3BP1 actually does not show size expansion under the influence of 1,6-HD, whereas FUS and other proteins do. This can be traced back to the electrostatic self-inhibitory confirmation of the protein, which is primarily affected by the ionic strength. Please refer to Response Figure 5.

Response Figure 5. Polypeptide chain expansion under low ionic strength conditions. (a) Hydrodynamic radius change comparison between FUS and G3BP1 at low ionic strength conditions. (b) Sequence charge fraction comparison between FUS and G3BP1 as well as their disordered regions. (c) Illustration of the G3BP1 IDR sequence charge as reproduced from ⁶ (top) and ⁷ (bot). (d) Hydrodynamic radius change of G3BP1 under change of the 1,6-HD concentration.

R3: Addressed. The authors included new experiments showing the effect of KCl concentration on hydrodynamic radius (in response Figure 5; note: I do not see this figure in the main or supplemental files, so perhaps it was strictly for the rebuttal).

9. Lines 173-175, the observation that ion release triggers phase separation is overstated. A correlation of these two states was observed, but causation was not determined. The sentence should be reworded, or additional direct measurements should be reported to support this claim.

We thank the reviewer for pointing this out and agree that the statement could benefit from rewording. Specifically, the decrease in ionic strength acts as a trigger for allowing phase separation to become spontaneously favourable. Here, ion exclusion from the dense phase and a release of ions from the polypeptide chains are significant contributors to overall driving force of phase separation. We have amended the phrasing accordingly (Lines 146 and 147).

R3: Addressed. The authors corrected the text accordingly.

10. Line 190, strong non-ionic interactions arising from an increase in ionic strength is mentioned. Could authors list, which non-ionic interactions they are considering here?

The driving forces for high salt reentrant phase separation have previously been shown in detail to include non-ionic interactions such as pi-stacking or non-polar contacts as well as hydrophobic interactions driven by interfacial water release⁵. The statement was expanded to provide more detail discussion (Line 173ff).

R3: Addressed. The authors corrected the text accordingly.

11. Line 326, the authors mention “a multitude of commonly available measurement assays”, but when introducing their method, they only list epi-fluorescence and absorbance. Are there any more? Please list. Alternatively, remove “multitude” as it evokes a high number of possible approaches.

We thank the reviewer for raising this comment and have changed ‘a multitude of commonly available measurement assays’ to ‘using commonly available measurement assays’ (Line 345)

R3: Addressed. The authors corrected the text accordingly.

12. The presented framework relies on a classic equilibrium thermodynamics. Have the authors

considered entropy and enthalpy connected with such phase separation and its possible effect in vivo?

Entropy and enthalpy effects are both contained in the general free energy $f(\phi)$ used in the dominance framework. Qualitatively, the entropic effect dominates at very low protein concentrations due to the logarithmic scaling of the chemical potential originating from translational entropy, and enthalpy is more important at higher protein concentrations due to its polynomial scaling. A detailed discussion of how entropy and enthalpy fit in the dominance framework requires introducing mathematical models of phase separation (Flory-Huggins for instance) and analysing their phase space using the dominance approach, which is quite beyond the scope of the current manuscript. In cells, active processes could potentially change this equilibrium picture, but it is possible to treat them as cases of dynamic equilibrium, so the proposed framework remains applicable.

R3: This question is more appropriate for a review and beyond the scope of the current study.

13. The applicability of these observations to in vivo systems is unclear, due to many obvious differences between the systems. Beside activity, etc, liquid condensates in vivo contain RNA as ubiquitous ingredient, with RNA being charged, flexible, etc, hence very likely influencing the effect of salts on proteins in condensates in vivo. Thus, it is difficult to draw conclusions about phase behavior of liquid condensates in vivo.

We appreciate the opportunity to discuss this work in the context of complex systems, such as within cells. As the reviewer rightly pointed out, the diversity of biomolecules and interactions introduces additional factors to consider. However, the fundamental physico-chemical principles driving protein interactions, such as the release of part of the ionic hydration shell to enable intermolecular interactions, are unlikely to be rendered completely irrelevant even in more complex systems.

That said, we agree with the reviewer that it is crucial to also study the phase behaviour of condensates in vivo. To address this, we reanalysed *in cellulo* datasets from the literature, demonstrating that our approach is applicable to complex environments. In doing so we can assess energetic differences in the collective interactions between opto-protein variants and G3BP1. For detailed discussion, please refer to Response Figure 2 below and manuscript p. 8, where we have added a new section titled 'Translation to complex environments' based on this data.

In summary, understanding the fundamental physico-chemical processes driving protein interactions is key to unravelling the complexities of intracellular environments. Our approach, as supported by the additional data, enables this characterisation directly in cells.

Response Figure 6. Intracellular dominance determination. (a) Homotypic line scans are performed by quantifying dilute phase concentrations of component A while varying its total concentration. (b) Homotypic response function of G3BP1 *in cellulo* from 1 . (c) Comparison of homotypic response functions of opto-protein fusion constructs intracellularly from 1 . (d) Dominance fraction comparison of different proteins in (c).

R3: Addressed. The authors included optogenetic clustering data to map FUS and G3BP1 phase behavior in cell and to calculate dominance fraction (new Figure 6).

Minor Points:

1. Line 25 typo: “would enable to address”

This has been corrected.

2. Figure 4e and 4f are discussed but not explicitly referenced in the text anywhere.

Figures 4e and 4f are now referred to in the text at the according positions (Lines 199 and 205).

3. Figure S3, subplots are not labelled, and referred to as “a”, “b”, and “left panel”.

This has been corrected.

4. Supplementary Figure S2 is labeled as S1, and all subsequent SI figure titles are off by 1.

We thank the reviewer for pointing this out, which has been amended accordingly.

5. Figure 4a: “reentrant phase transition”

Corrected.

6. Figure 4c: “no-ionic interactions” should be “non-ionic interactions.”

Yes, correct – amended. Thank you.

R3: All minor points have been addressed

References

1. Riback, J. A. *et al.* Composition-dependent thermodynamics of intracellular phase separation. *Nature* **581**, 209–214 (2020).
2. Welsh, T. J. *et al.* Surface Electrostatics Govern the Emulsion Stability of Biomolecular Condensates. *Nano Lett* **22**, 612–621 (2022).
3. Li, L. *et al.* Phase Behavior and Salt Partitioning in Polyelectrolyte Complex Coacervates. *Macromolecules* **51**, 2988–2995 (2018).
4. Friedowitz, S. *et al.* Looping-in complexation and ion partitioning in nonstoichiometric polyelectrolyte mixtures. *Science Advances* **7**, eabg8654.
5. Krainer, G. *et al.* Reentrant liquid condensate phase of proteins is stabilized by hydrophobic and non-ionic interactions. *Nature Communications* **12**, 1085 (2021).
6. Guillén-Boixet, J. *et al.* RNA-Induced Conformational Switching and Clustering of G3BP Drive Stress Granule Assembly by Condensation. *Cell* **181**, 346–361.e17 (2020).
7. Yang, P. *et al.* G3BP1 Is a Tunable Switch that Triggers Phase Separation to Assemble Stress Granules. *Cell* **181**, 325–345.e28 (2020).
8. André, A. A. M., Yewdall, N. A. & Spruijt, E. Crowding-induced phase separation and gelling by co-condensation of PEG in NPM1-rRNA condensates. *Biophysical Journal* **122**, 397–407 (2023).
9. Qian, D. *et al.* Tie-lines reveal interactions driving heteromolecular condensate formation. *bioRxiv* 2022.02.22.481401 (2022) doi:10.1101/2022.02.22.481401.